# Characterization of a new type of neuronal 5-HT G- protein coupled receptor in the cestode nervous system

Federico Camicia[1], Hugo R. Vaca[2,3], Sang-Kyu Park[4], Augusto E. Bivona[2,3], Ariel Naidich[5], Matias Preza[6], Uriel Koziol[6], Ana M. Celentano[2,3], Jonathan S. Marchant[4]*, Mara C. Rosenzvit[2,3]*

**1** Laboratorio de Toxinopatología, Centro de Patología Experimental y Aplicada, Facultad de Medicina, Universidad de Buenos Aires (UBA), Ciudad Autónoma de Buenos Aires, Argentina, **2** Instituto de Investigaciones en Microbiología y Parasitología Médica (IMPaM, UBA-CONICET), Facultad de Medicina (FMed), Universidad de Buenos Aires (UBA), Consejo Nacional de Investigaciones Científicas y Técnicas (CONICET), Ciudad Autónoma de Buenos Aires, Argentina, **3** Departamento de Microbiología, Parasitología e Inmunología, Facultad de Medicina, Universidad de Buenos Aires (UBA), Ciudad Autónoma de Buenos Aires, Argentina, **4** Department of Cell Biology, Neurobiology & Anatomy, Medical College of Wisconsin, Milwaukee, WI, United States of America, **5** Departamento de Parasitología, INEI-ANLIS, "Dr Carlos G. Malbrán", Ciudad Autónoma de Buenos Aires, Argentina, **6** Sección Biología Celular, Facultad de Ciencias, Universidad de la República, Montevideo, Uruguay

* jmarchant@mcw.edu (JSM); mrosenzvit@fmed.uba.ar (MCR)

**Data Availability Statement:** All the accession numbers are available from the Genbank database (accession number(s) MW535743 for Eca-5HT1a;

## Abstract

Cestodes are platyhelminth parasites with a wide range of hosts that cause neglected diseases. Neurotransmitter signaling is of critical importance for these parasites which lack circulatory, respiratory and digestive systems. For example, serotonin (5-HT) and serotonergic G-protein coupled receptors (5-HT GPCRs) play major roles in cestode motility, development and reproduction. In previous work, we deorphanized a group of 5-HT7 type GPCRs from cestodes. However, little is known about another type of 5-HT GPCR, the 5-HT1 clade, which has been studied in several invertebrate phyla but not in platyhelminthes. Three putative 5-HT GPCRs from *Echinococcus canadensis*, *Mesocestoides vogae* (*syn. M. corti*) and *Hymenolepis microstoma* were cloned, sequenced and bioinformatically analyzed. Evidence grouped these new sequences within the 5-HT1 clade of GPCRs but differences in highly conserved GPCR motifs were observed. Transcriptomic analysis, heterologous expression and immunolocalization studies were performed to characterize the *E. canadensis* receptor, called Eca-5-HT$_{1a}$. Functional heterologous expression studies showed that Eca-5-HT$_{1a}$ is highly specific for serotonin. 5-Methoxytryptamine and α-methylserotonin, both known 5-HT GPCR agonists, give stimulatory responses whereas methysergide, a known 5-HT GPCR ligand, give an antagonist response in Eca-5-HT$_{1a}$. Mutants obtained by the substitution of key predicted residues resulted in severe impairment of receptor activity, confirming that indeed, these residues have important roles in receptor function. Immunolocalization studies on the protoscolex stage from *E. canadensis*, showed that Eca-5-HT$_{1a}$ is localized in branched fibers which correspond to the nervous system of the parasite. The patterns of immunoreactive fibers for Eca-5-HT$_{1a}$ and for serotonin were intimately intertwined but not identical, suggesting that they are two separate groups of

MW535744 for Mvo-5HT1a and MW535745 for Hmi-5HT1a).

**Funding:** This work was supported by Consejo Nacional de Investigaciones Científicas y Técnicas (CONICET), Argentina and by Secretaria de Ciencia y Técnica (UBACyT), Universidad de Buenos Aires, Facultad de Medicina, Argentina. Project Programación Científica 2016, code 20020150100160BA, PICT 2017 N°2966. HRV is a recipient of a CONICET postdoctoral fellowship. JSM and SKP were supported by the NIH (R01 AI145871). The funders had no role in study design, data collection and analysis, decision to publish, or preparation of the manuscript.

**Competing interests:** The authors have declared that no competing interests exist.

fibers. These data provide the first functional, pharmacological and localization report of a serotonergic receptor that putatively belongs to the 5-HT1 type of GPCRs in cestodes. The serotonergic GPCR characterized here may represent a new target for antiparasitic intervention.

## Introduction

The parasitic flatworms *Echinococcus granulosus sensu lato* (*s.l.*), *Hymenolepis microstoma* and *Mesocestoides vogae* (*syn. M. corti*) are tapeworms belonging to different families of the class Cestoda, with *E. granulosus s.l.* belonging to Taeniidae, *H. microstoma* to Hymenolepididae and *M. vogae* to Mesocestoididae family. Several species of the genus *Echinococcus* cause parasitic diseases in wildlife, domestic animals and humans worldwide. The larval stage of almost all species of the *E. granulosus s.l.* complex [which includes *E. granulosus sensu stricto* (*s.s.*) and *E. canadensis*] cause human cystic echinococcosis (formerly hydatidosis) [1], one of the 17 neglected diseases prioritized by WHO [2]. Parasites of the genus *Hymenolepis* are known to develop, depending on the species, in humans, rats and/or mice. *H. nana* is a highly prevalent human intestinal tapeworm, especially affecting children [3]. *H. microstoma*, the mouse bile duct tapeworm, is a rodent/beetle-hosted laboratory parasite that has been used as a model for the study of cestodes [4]. In recent years, the genome of *H. microstoma* has been sequenced and assembled [5]. *M. vogae* is a cestode that has a complex life cycle with arthropods as first intermediate hosts, mouse or lizards as second intermediate hosts and carnivores (dogs, cats or skunks) that host the adult intestinal tapeworms [6]. This parasite is a well-established model for laboratory studies [7] and is easy to maintain and reproduce. The larval stage (tetrathyridia) of *M. vogae* has a remarkable capacity of asexual reproduction in the peritoneal cavity of mice and some other mammalian hosts [8]. The tetrathyridium is used to examine drug effects on neuromuscular activity [9, 10].

The helminth nervous system plays essential functions in motility, chemosensation, reproduction, attachment, and feeding [11]. The importance of neuronal signaling events makes its component elements attractive targets for chemotherapy. For example, ligand-gated ion channels are already employed as drug targets in nematode treatment. G-protein coupled receptors (GPCRs) are also good examples of druggable targets for new compounds [11], as exemplified by the action of ligands that modulate the serotonergic system in flatworms [11]. The ability of anthelmintics to bind to GPCRs is supported by the finding that the antischistosomal drug praziquantel functions as a partial agonist of a human serotonergic GPCR [12]. In mammals, seven major types of 5-HT receptors are known but in invertebrates, only three types or clades have been found [13]. The invertebrate receptors are classified according to homology with mammalian receptor families as 5-HT1, 5-HT2 and 5-HT7-like, being the classified system followed in this work. 5-HT1 type activated receptors are preferentially coupled with the alpha subunit of an inhibitory G-protein ($G\alpha_{i/o}$) which inhibits adenylate cyclase [13]. Activation of 5-HT2 type receptors are usually characterized by a rise in cytoplasmic calcium following the activation of $G\alpha_{q/11}$ [13]. Finally, the 5-HT7 type, include receptors which are preferentially coupled with $G\alpha_s$ leading to the activation of adenylate cyclase and an increase in intracellular cyclic AMP [13]. In previous works, several serotonergic GPCRs were identified in cestodes [5, 14]. However, at the time of writing, all the serotonergic GPCRs functionally reported from platyhelminths, are grouped within type 7 serotonergic GPCRs [10, 15–18]. Genome searches predicted the presence of additional 5-HT GPCRs in platyhelminths [15, 16]. In nematodes, it

is well known that 5-HT1 type receptors play major roles in motor control [19] and in some parasitic nematode species it has been shown that 5-HT1 agonists could be used as anthelmintic agents [20]. In recent years, Wang & coworkers [21] reported the crystal structures of the human (*Homo sapiens*) Hsa-5-HT$_{1b}$ GPCR when bound to the ergot alkaloids ergotamine and dihydroergotamine. These drug bound structures reveal the important roles of critical motifs and specific amino acid residues in transmembrane domains 3 and 5 (besides others) that form the orthosteric pocket or ligand binding site of the receptor. Another important structural breakthrough was the report of the first molecular structure of the serotonin Hsa-5-HT$_{1b}$ receptor coupled to heterotrimeric G$\alpha_o$ [22].

In this work, we report for the first time, the cloning, sequencing and bioinformatic characterization of a new 5-HT GPCR type, that likely belongs to the 5-HT1 class (clade 4), in several species of cestodes. We functionally characterize and localize one of these 5-HT1 type GPCRs from *E. canadensis* which shows a particular pharmacology. Moreover, mutant analyses seem to confirm pivotal roles played by some residues located at specific positions in the receptor´s sequence. Our results demonstrate that this GPCR is present in the cestode nervous system, where it may play a role in neuromuscular function, making it a viable target for future chemotherapeutic intervention.

## Material and methods

### Ethics statement

Experiments involving the use of experimental animals were carried out according to protocols approved by the Comité Institucional para el Cuidado y Uso de Animales de Laboratorio (CICUAL), Facultad de Medicina, Universidad de Buenos Aires, Argentina (protocol "Farmacología, localización y función de receptores acoplados a proteínas G (GPCRs) y de canales de calcio de *Echinococcus granulosus* y otros parásitos cestodes como posibles blancos de drogas antiparasitarias" CD N˚ 2542/2019). For *M. vogae* passages, the protocol "Pasaje in vivo de parásitos cestodes de la especie *Mesocestoides corti*" CD N˚1127/2015 and 1229/2015 was followed. Cyst puncture was performed following the approved protocol by the same institution (protocol "Punción de quistes hidatídicos de infecciones naturales" CD N˚ 3723/2014). *H. microstoma* passages were performed according to the protocol "Mantenimiento del ciclo vital completo del cestodo *Hymenolepis microstoma* utilizando sus hospedadores naturales *Mus musculus* (ratón) y *Tribolium confusum* (escarabajo de la harina)"; Protocol number: 10190000025215, Comisión Honoraria de Experimentación Animal, Uruguay.

### Chemicals

5-HT (H9523), α-methylserotonin (M110), 5-methoxytryptamine (286583), tryptamine (193747), tyramine (T90344), octopamine (O0250), acetylcholine (A6625), histamine (H7125), dopamine (H8502) and methysergide (M137) were obtained from Sigma Chemical Company. Tryptamine, α-methylserotonin, 5-methoxytryptamine and methysergide were dissolved in DMSO whereas tyramine, octopamine, acetylcholine, histamine and dopamine were dissolved in distilled H$_2$O, all of them (with the exception of 5-HT) at 100 mM. 5-HT was dissolved in distilled H$_2$O at stock concentration of 5 mM. Stock solutions were either made up on the day of the experiment or taken from aliquots stored at -80˚C for no longer than 1 week prior to use. After 0.22 μm filtration with Millex GV filter units (Millipore, Ireland), stock solutions were diluted to the corresponding final concentration (e.g. 0.0001; 0.001; 0.01; 0.1; 1; 10 and 50 μM for 5-HT) in RPMI medium with high glucose (Gibco, USA).

## Animals

Wistar female rats and BALB/c female mice were bred and housed at the animal facilities of Instituto de Investigaciones en Microbiología y Parasitología Médica (IMPaM), Facultad de Medicina, Universidad de Buenos Aires (UBA)-Consejo Nacional de Investigaciones Científicas y Técnicas (CONICET), Buenos Aires, Argentina, in a temperature-controlled light cycle room with food and water *ad libitum*. C57BL/6 female mice were bred and housed in collaboration with Jenny Saldaña and Laura Domínguez, Laboratorio de Experimentación Animal, Facultad de Química, Universidad de la República, Uruguay. Experiments with animals were approved by the Review Board of Ethics of the School of Medicine (UBA, Argentina) and conducted in accordance with the guidelines established by the National Research Council.

## Parasite material

*E. granulosus s.l.* protoscoleces were obtained under sterile conditions by needle aspiration of hepatic hydatid cysts of porcine origin. Parasite material was kindly donated by Alimentaria La Pompeya, Marcos Paz, Provincia de Buenos Aires, Argentina. The livers used for parasite extraction were from animals that were not specifically used for this study and all the material obtained was processed as part of the normal work of the abattoir. Samples from animals at the abattoir were collected under consent from local authorities. Protoscolex viability was assessed using the eosin exclusion test after three washes with PBS, with 50 μg/ml of gentamicin to remove cyst wall debris [23]. Only samples showing more than 95% viability were used. A fraction of the protoscoleces was used for whole mount immunohistochemistry, another fraction was used for cDNA synthesis and the remaining protoscoleces were used for species/genotype determination by sequencing a fragment of the mitochondrial cytochrome c oxidase subunit 1 (CO1), as previously described [24]. The resulting species and genotype of all protoscoleces used in this work were from *E. canadensis* G7. Three biological replicates were used with each replicate corresponding to protoscoleces obtained from a single cyst. *M. vogae* larvae (tetrathyridia) were maintained by alternate, serial passages in Wistar female rats and BALB/c female mice as previously described [6]. The tetrathyridia larvae were obtained from the mouse intraperitoneal cavity after 3 months of intraperitoneal inoculation. Only tetrathyridia from up to the third serial passage in mice were used for the experiments. The life cycle of *H. microstoma* was maintained using C57BL/6 mice as definitive hosts, and *Tribolium confusum* beetles as intermediate hosts [4]. Adults were obtained from the bile duct of infected mice after 3 to 4 months post-infection.

## Bioinformatic analyses

The selection of the sequences for the cloning of *E. canadensis*, *H. microstoma* and *M. vogae* 5-HT1 GPCRs were based on bioinformatic analysis done by our group previously to this work [5, 10, 14]. Transmembrane segments were predicted using the topology prediction server HMMTOP (http://www.enzim.hu/hmmtop/index.php) and bidimensional models of receptors were generated using Protter (http://www.enzim.hu/hmmtop/index.php). Analyses of critical residues for serotonergic GPCR ligand binding, G-protein coupling and clade characterization were performed using only platyhelminth sequences from different serotonergic clades and aligned using ClustalW program from the expasy proteomic package (https://embnet.vital-it.ch/software/ClustalW.html). The Ballesteros and Weinstein (B&W) nomenclature for residue numbering was followed here [25]. For the search of best orthologues, blast searches were performed in wormbase parasite (https://parasite.wormbase.org/index.html) and only sequences with at least 70% of amino acid identity were chosen for the alignment. Only some gene models were chosen for the alignment, which also gave the best alignments by

visual inspection in multiple sequence alignments. The protein sequence alignments were performed using ClustalW alignment utility integrated to MEGA X software [26–28]. Trees were built according to the Neighbour-Joining method using the Best Tree mode available in MEGA X and were verified by bootstrap analysis with 500 replicates. Cloned and sequenced Eca-5-HT$_{1a}$, Mvo-5-HT$_{1a}$ and Hmi-5-HT$_{1a}$ were aligned with serotonin receptor sequences from the following invertebrates: *E. canadensis*, *M. vogae*, *Dugesia japonica*, *Schistosoma mansoni*, *Caenorhabditis elegans* and *Drosophila melanogaster*. With the exception of most of the sequences from *D. japonica* and *S. mansoni*, the rest of the sequences used for Phylogenetic groupings and multiple sequence alignments in tables were from receptors functionally tested and with G- protein coupling partners determined. The tree drawing was obtained using Fig-Tree v 1.4.4 [29]. For prediction of coupling specificity, the web server program PRED COUPLE 2.00 was used (http://bioinformatics.biol.uoa.gr/PRED-COUPLE) [30, 31]. The predictive capacity of the program was previously tested using ten sequences from serotonergic receptors belonging to different invertebrate species for which coupling specificity was experimentally determined (S1 Table). In all the cases tested, we found a perfect match between the G-protein coupling partner experimentally determined and the predicted by the program.

## Levels of GPCR expression

The transcriptional expression levels (in RPKM, or reads per kilobase per million reads) for each serotonergic receptor in *E. granulosus s.s.* (G1 genotype) were from Zheng et al. [32].

## Homology modeling

For molecular homology modeling studies, protein domains of the 5-HT1-type receptors studied in this work (Eca-5-HT$_{1a}$, Mvo-5-HT$_{1a}$ and Hmi-5-HT$_{1a}$), the 5-HT7-type receptors Eca-5-HT$_{7a}$ and Mvo-5-HT$_{7a}$, studied previously [10], and the 5-HT7-type receptor from *H. microstoma* (Hmi-5-HT$_{7a}$; Wormbase parasite gene ID HmN_000156500) were screened against PFAM databases, using PfamScan [33]. Protein structure homology models were performed using PHYRE2 [34] and SWISS-MODEL [35–37] databases. For all sequences, already crystallized structures from *H. sapiens* (PDB IDs: 4IAQ and 4IAR) [21] were used as templates for modeling seven transmembrane domains. The intracellular loop 3 (ICL3) and C-terminus (C-term) regions were modeled *ab initio* using PHYRE2. The homology model for the human receptors Hsa-5-HT$_{1a}$ (P08908) and Hsa-5-HT$_{7a}$ (P34969) were also performed. The molecular visualization and figures generated in this work were performed using the software PyMOL version 2.0.4 (https://pymol.org). All homology models obtained were validated calculating several parameters such as: ERRAT [38] (https://servicesn.mbi.ucla.edu/ERRAT/), QMEN and Ramachandran plots, which were calculated using Structure Assessment Tool of SWISS-MODEL (https://swissmodel.expasy.org/assess). RMSD and TM-align [39] (https://zhanglab.ccmb.med.umich.edu/TM-align/) were determined to compare the homology models to Hsa-5-HT$_{1b}$ (PDB ID: 4IAR). Finally, structural comparisons were performed to identify relevant and conserved residues in the ligand binding site and G- protein interaction site.

## RNA extraction and cDNA synthesis

Total RNA from *E. canadensis* was isolated from protoscoleces that were crushed under liquid nitrogen and processed using Trizol reagent (Invitrogen). The RNA obtained was treated with RNase-Free DNase (Fermentas), ethanol precipitated and reverse transcribed using Superscript III reverse transcriptase (RT) (Invitrogen) and gene specific reverse primer complementary to 5-HT GPCR gene models. One cDNA for each selected gene model was synthesized. The same procedure was followed to obtain RNA from *M. vogae* tetrathyridia and cDNA

synthesis. Total RNA from *H. microstoma* was isolated from adult stage obtained from the bile ducts of C57BL/6 mice and purified with TRI Reagent (Sigma Aldrich) in combination with the Direct-zol RNA Miniprep (Zymo Research). Reverse transcription was performed with Superscript II (Invitrogen) according to the instructions of the manufacturer.

## Amplification and cloning of cDNAs coding for serotonergic GPCRs

The gene model called EgrG_001050800.1 from *E. granulosus s.s.* was first identified by reciprocal BLAST searches [14] from the published genome of *E. granulosus* [5]. Using primers against the contiguous regions to the ends of the coding region of this receptor-coding gene and employing cDNA from *E. canadensis*, the full-length cloning of this cDNA was accomplished. The sequence of the forward primer was `5´ CAGCGTGCTGAGGTGCCAATC 3´` and that of the reverse primer was `5´ GAGTTGTGGGGACAAGGTTGC 3´`. The product obtained by PCR amplification using Q5® High-Fidelity DNA Polymerase (New England Biolabs) had the expected size (~1900 base pairs, bp). The cycling parameters for the PCR were: initial denaturing step 98 degrees, 30 seconds followed by 35 cycles of the following steps: melting, 98˚C, 10 seconds; annealing 66˚C, 30 seconds and extension, 72˚C, 2,5 minutes; the amplification finished with a final extension step of 30 minutes at 72˚C. For *M. vogae*, the gene model called MCU_000790-RA was identified by BLAST searches of the predicted proteome of this species [40] using the sequence from *E. canadensis* as a bait. Using primers against the contiguous regions to the ends of the coding region of this receptor and employing cDNA from *M. vogae*, the full-length cloning was later accomplished. The sequence of the forward primer was `5´ GATTAATGCCTCCACTCACAC 3´` and that of the reverse primer was `5´ GTCTGCTTTCGTC ACTTAAAGTAG 3´`. Using these primers in PCR amplification with Q5® High-Fidelity DNA Polymerase (New England Biolabs) an amplification product of the expected size (~1900 bp) was obtained. The cycling parameters for the PCR were: initial denaturing step 98˚C, 30 seconds; followed by 35 cycles of the following steps: melting, 98˚C, 10 seconds; annealing 60˚C, 30 seconds and extension, 72˚C, 2,5 minutes; the amplification finished with a final extension step of 30 minutes at 72˚C. For *H. microstoma*, the gene model called HmN_000578900 was identified by BLAST searches using the sequence from *E. canadensis* as a bait. The forward primer sequence was `5-ATGGCGGCCACAGTTTCTC-3` and the reverse primer was `5-ATC GAAAACAATTCAATCGTG-3`. Both primers gave an expected product size (1824 bp) by PCR amplification using High Taq DNA polymerase (Bioron, Germany). The cycling parameters for the PCR were: initial denaturation 94˚C, 2 minutes; followed by 30 cycles of the following steps: melting, 94˚C, 10 seconds; annealing, 52˚C, 20 seconds; elongation, 72˚C, 2 minutes. The protocol finished with a final extension step at 72˚C, 30 minutes. Amplification products were visualized by agarose gel electrophoresis and Gel Red staining and the bands of interest were extracted from the gel using the QIAquick Gel Extraction Kit (Qiagen) and cloned into the pTOP Blunt cloning vector (Macrogen). The recombinant plasmids were used for *Escherichia coli* (DH5α) transformation and the transformed bacteria were grown in LB medium with ampicillin and kanamycin. The selected colonies were then used for plasmid purification using the GeneJet Plasmid miniprep kit (Fermentas) and sequencing using an Applied Biosystems Big Dye terminator kit (Applied Biosystems) on an ABI 377 automated DNA sequencer. The cloned cDNAs from *E. canadensis*, *M. vogae* and *Hymenolepis microstoma* were named Eca-5-HT$_{1a}$, Mvo-5-HT$_{1a}$ and Hmi-5-HT$_{1a}$ respectively.

## Fluorescence Imaging Plate Reader (FLIPR) assay

HEK293 cells (ATCC CRL-1573.3) or Gα$_{15}$ stably transfected HEK293 cells (HEK293-Gα$_{15}$) were cultured in growth media [DMEM (Gibco), 10% heat inactivated fetal bovine serum

(Gibco), penicillin (100 units/mL), streptomycin (100 μg/mL) and L-glutamine (290μg/mL)] and used for assays between passages 5 and 25. For cestode GPCR heterologous expression assays, cells were transfected (Lipofectamine 2000, Invitrogen) at 80% confluency approximately 16 hours after seeding within 60 mm plate dish with 6 μg of plasmid DNA of human codon optimized cestode GPCR cDNA (subcloned into a pcDNA3.1(-) mammalian expression vector). The following day, cells were trypsinized, centrifuged (300g/5min), resuspended in DMEM supplemented with 1% dialyzed FBS (Gibco) and plated in 96 well, black-walled clear-bottomed poly-d-lysine coated 96-well plate (Corning). Cells were seeded (50,000 cells/well) in DMEM growth media with 10% dialyzed FBS and allowed to adhere. After 24 hours, growth medium was removed, and 100 μl of 1× Fluo-4 NW dye loading solution (Invitrogen), which was reconstituted with an assay buffer containing 2.5 mM probenecid in 1× Hanks' balanced salt solution (HBSS) with $Ca^{2+}$, $Mg^{2+}$, and 20 mM HEPES was added to each well. The cells were incubated for 30 min at 37°C in 5% $CO_2$ in air followed by an additional 30-min incubation at room temperature. Dilutions for each drug were prepared in RPMI medium, without probenecid and dye, in V-shape 96-well plates (Greiner Bio-one, Germany). The calcium assay was performed at room temperature using a FLIPR$^{TETRA}$ (Molecular Devices). Basal fluorescence was monitored for 20 s, then 25 μl of each drug was added, and the signal was monitored over 250 seconds. Results were expressed as relative fluorescence units (RFU). At the end of $[Ca^{2+}]_i$ measurements, cells were examined for morphological changes. Data analyses were performed using Excel and Origin to yield the EC50 parameter (half maximal effective concentration). Controls consisted of mock transfected HEK293-G$\alpha_{15}$ cells and HEK293 cells.

### Design of mutants for Eca-5-HT1a

Four different plasmid constructs were designed by request in Genscript company replacing native residues from Eca-5-HT$_{1a}$ by alanine. The mutants designed were: glutamate replaced by alanine (D118A), cysteine by alanine (C122A), threonine by alanine (T123A) and finally tryptophan by alanine (W542A). Each of the four mutants for cestode GPCR was subcloned into a pcDNA3.1(-) mammalian expression vector and used for HEK293-G$\alpha_{15}$ cell transfection. Mutants and wild type Eca-5-HT$_{1a}$ calcium responses were compared. Data analyses were performed using Excel and Origin to yield the EC50 parameter (half maximal effective concentration).

### Cell transfection with Eca-5-HT$_{1a}$ and cell line generation

HEK293 cells ($1 \times 10^6$) stably expressing GNA15 (HEK-GNA15) were seeded into 60 mm dishes and after 24 hours were transfected with 5 μg of plasmid DNA, pCMV6-A-Puro Eca-5-HT$_{1a}$ (Eurofins Blue Heron Biotech) by using Lipofectamine 2000 reagent. Forty-eight hours after transfection, cells were trypsinized and seeded into 100-mm dishes and selected by 200 μg/ml hygromycin and 2 μg/ml puromycin for 7–10 days. Single colonies were isolated, and the expression of Eca-5-HT$_{1a}$ was examined by FLIPR$^{TETRA}$.

### Bacterial expression of the third intracellular loop of Eca-5-HT$_{1a}$ and preparation of polyclonal antibodies

The sequence from the nucleotide position 673 to 1581 of the cDNA that encodes for a protein fragment of 303 amino acids which represent the entire ICL3 of Eca-5-HT$_{1a}$ receptor was codon optimized and the nucleotide fragment synthesized (Eca-5-HT$_{1a}$ICL3, S1 Text). The synthesized sequence was cloned into the vector pET-30a (+) with His tag for protein expression. *E. coli* strain BL21 star (DE3) was transformed with the recombinant plasmid mentioned above. Recombinant BL21 star (DE3) stock stored in glycerol was thawed and inoculated in 4

ml LB medium containing 50 μg/ml kanamycin and incubated overnight at 37˚C with shaking at 200 rpm. The 4 ml pre-culture was seeded into 2 X 500 ml Terrific Broth Liquid microbial growth medium containing 50 μg/ml kanamycin in a 2 liters Erlenmeyer flask, and incubated at 37˚C with shaking at 200 rpm. When the OD600 value of the culture reached 1.2, IPTG was added at final concentration of 0.5 mM to induce protein expression at 15˚C for 16h with shaking at 200 rpm. Cells were harvested by centrifugation at 8,000 g at 4˚C for 20 min. Cell pellets were resuspended with lysis buffer (50 mM Tris-HCl,150 mM NaCl, 1mM DTT, pH 8.0) followed by sonication cycles consisting of alternating 3 seconds on and 6 seconds off for a total of 6 min at 600 w. The precipitate after centrifugation was dissolved using a denaturing agent (50 mM Tris-HCl, 8M Urea, 1mM DTT, pH 8.0). Eca-5-HT$_{1a}$ICL3 protein was obtained by one-step purification using Ni$^{2+}$ column and eluted with elution buffer (50 mM Tris-HCl, 8M Urea, 500 mM Imidazole, pH 8.0). Target protein was pooled and refolded by dialyzing into buffer PBS, 10% Glycerol, 0.2% SDS, pH 7.4. After dialysis, the sample was filtered through a 0.22 μm filter. The concentration was determined by BCA$^{TM}$ protein assay (ThermoFisher, Cat. No. 23225) with bovine serum albumin as standard. The protein purity was around 85% as estimated by densitometric analysis of the Coomassie Blue-stained SDS-PAGE gel under reducing conditions. The molecular weight observed was approximately 40 kDa and close to the predicted molecular weight, as determined by standard SDS-PAGE along with Western blot confirmation (S1A Fig). Western blots were performed using "THE™ His-Tag Antibody, mAb, Mouse" as primary antibody (GenScript, Cat. No. A00186) and revealed using rabbit anti-mouse IgG H&L (HRP) (ab6728) as secondary antibody. Six-week-old female Balb/c mice (n = 6) were immunized four times (on days 0, 10, 20 and 30) with 20 μg of the recombinant Eca-5-HT$_{1a}$ICL3 (S1 Text) by intraperitoneal injection. Priming dose was administered with complete Freund's adjuvant (Sigma-Aldrich) and subsequent doses were formulated with incomplete Freund's adjuvant (Sigma-Aldrich). Fifteen days after the last immunization, mice were exsanguinated under general anesthesia and then euthanized by cervical dislocation. Blood was allowed to coagulate at 37˚C for 30 min and finally serum was separated by centrifugation at 2000 g for 10 min. Anti-Eca-5-HT$_{1a}$ titers in serum were determined by ELISA (S1B Fig). Briefly, a 96-well flat bottom polystyrene plate (Nunc, Thermo Scientific) was coated with 0.2 μg of recombinant Eca-5-HT$_{1a}$ICL3 per well O.N. at 4˚C. Then, non-specific binding sites were blocked with a solution of 3% BSA in PBS for 2 h at 37˚C. After washing the plate three times with 0.05% Tween-20 solution, 2-fold serial dilution of mice sera were incubated O.N. at 4˚C (pre-immune serum was included as negative control). Next day, the plate was washed again and an anti-IgG-HRP (Sigma B6398) antibody (1/10000 dilution) was added as secondary antibody and incubated for 1 h at 37˚C. After washing again, the assay was revealed with TMB (tetramethylbenzidine, BD OptEIA™) and the reaction was stopped 15 min later with 4 N H$_2$SO$_4$. Absorbance at 450 nm was determined on an ELISA plate reader (Labsystems Multiscan EX). Specific antibody end-point titer was calculated as the reciprocal of the dilution with an Absorbance 450nm = 0.5.

## Whole-mount immunohistochemistry

Freshly obtained protoscoleces of *E. canadensis* from porcine hydatid cysts were activated by treatment with pepsin in DMEM (0.05% W/V) under acidic conditions (pH = 2) for 1 hour at 37˚C with shaking (125 rpm). Then, they were allowed to sediment, washed three times with PBS and treated with sodium taurocholate (0,2% W/V, checking that the pH remains at 7.4) for three hours at 37˚C with shaking (125 rpm). Finally, the protoscoleces were washed again 3 times with PBS. After washing, the viability of protoscoleces was checked with the eosin exclusion test and, if they had more than 95% viability, they were fixed overnight with 4%

paraformaldehyde at 4°C with shaking. After fixation, protoscoleces were washed three times with PBS-T at room temperature with shaking. The Immunohistofluorescence protocol of Koziol [41] for whole-mounts (WMIHF) without proteinase K treatment was then followed with minor modifications. Antibodies were used at the following dilutions: 1/50 for anti-Eca-5-HT$_{1a}$, 1/100 for anti-5-HT produced in rabbit (IgG whole molecule, catalog number S5545, Sigma) and 1/600 for anti-tropomyosin antibody produced in rabbit [42]. This latter antibody was a kind gift of Dr Gabriela Alvite, Sección Bioquímica, Universidad de la República, Uruguay. Secondary antibodies were IgG (H+L) Goat anti-Rabbit conjugated to FITC from Invitrogen™ and Goat anti-Mouse IgG (H+L) Secondary Antibody conjugated to Rhodamine, from ThermoFisher Scientific™ and both antibodies were used at 1/200. Specimens were finally mounted in a mix of 80% glycerol and 50 mM Tris-HCl, pH 8. Samples were viewed by a Spinning Disk-TIRF-Olympus-IX83 motorized microscope with confocal module (Disk Spinning Unit) coupled to a Hamamatsu Orca Flash 4 digital camera of 16 bits and a confocal Zeiss LSM 880 microscope with Airyscan detector for super-resolution. Images were captured at the best resolution possible (2048 x 2048). Images and stacks were viewed and processed using FIJI software (version 2.0.0-rc-69/1.52p).

## Results

### A new type of cestode serotonergic GPCR was cloned and bioinformatically analyzed

Based on previous bioinformatics analyses by our group [5, 10, 14], we cloned and sequenced a putative 5-HT1 GPCR receptor from *E. canadensis*. The cloned sequence was highly similar (~98% identity) but not identical to gene model EcG7_02049 from the published genome of *E. canadensis* [43] and was 1866 bp long with an open reading frame of 1821 bp, which encode a hypothetical receptor that was called here Eca-5-HT$_{1a}$.

According to the gene model (EcG7_02049) available in the Wormbase parasite database (https://parasite.wormbase.org/index.html), the gene sequence has three exons separated by two alternating introns. According to transcriptomic data available from the genome project of Zheng and coworkers [32], the messenger encoding for the *E. granulosus s.s.* orthologue of this receptor is highly expressed and restricted to the protoscolex stage (S2 Fig). Multiple sequence alignment with other orthologous sequences strongly suggested that the sequence found was complete. The predicted amino acid sequence is 606 residues long with a deduced molecular weight of 68,9 kDa. A program used to predict N-linked glycosylation (Protter) marked one potential site of N-linked glycosylation at the N-terminal end (Fig 1A). The Eca-5-HT$_{1a}$ receptor was highly conserved in other cestode species. Hypothetical orthologues of this receptor were also cloned and sequenced in *M. vogae* and *H. microstoma* (Fig 1B). The three cloned receptors grouped within the 5-HT1 clade (Fig 1C) were named as "Eca-5-HT$_{1a}$", "Mvo-5-HT$_{1a}$" and "Hmi-5-HT$_{1a}$" from serotonergic type 1 receptors from *E. canadensis*, *M. vogae* and *H. microstoma* respectively.

The predicted topology of these three cestode receptors comprised the canonical fold of seven transmembrane domains (7TM), with an extracellular amino terminal end, an intracellular carboxy terminal end and three alternating intracellular (ICLs) and extracellular (ECLs) loops (Fig 1A). The cestode sequences obtained here were aligned with hypothetical orthologues found in several plathyhelminth species, most of them in cestodes (Fig 1B).

This analysis suggests a strong degree of conservation of this hypothetical receptor in several plathyhelminth species. Many residues involved in ligand binding, receptor activation and receptor G-protein coupling are conserved (Fig 1B). The new cestode receptors grouped within the 5-HT1 clade of serotonergic receptors in the phylogenetic tree (Fig 1C). This type of

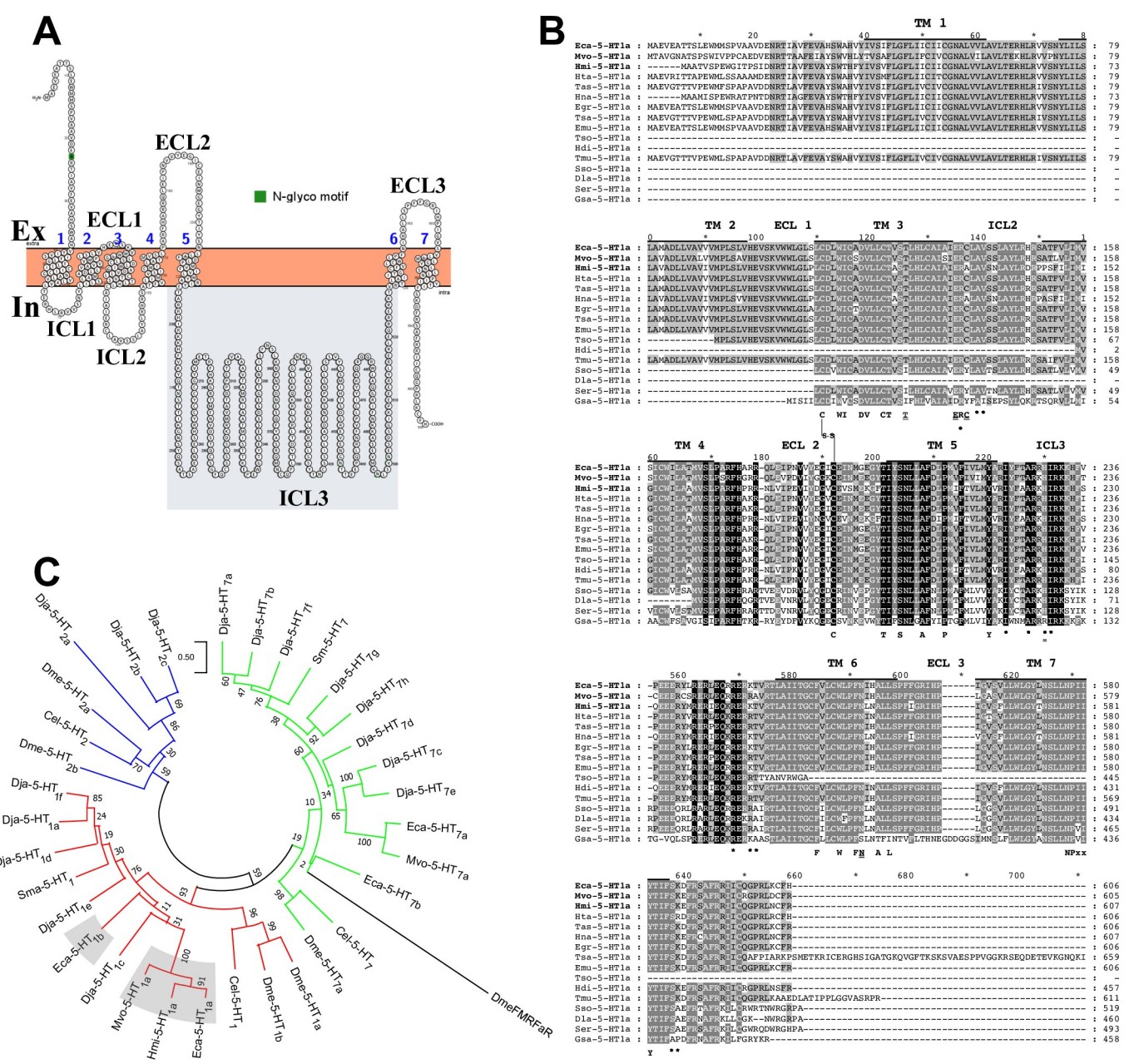

**Fig 1. Bioinformatic analyses of the three 5-HT1 serotonin receptors identified in this work (Eca-5-HT$_{1a}$, Mvo-5-HT$_{1a}$ and Hmi-5-HT$_{1a}$, GenBank accession numbers MW535743, MW535744 and MW535745 respectively).** A. The structural representation and prediction of residues of potential N-glycosylation of Eca-5-HT$_{1a}$ were obtained with the Protter program (http://www.enzim.hu/hmmtop/index.php). The intracellular and extracellular loops are indicated as ICL and ECL respectively. The third intracellular loop used for antibody generation is shaded in grey. Residues potentially involved in N-linked glycosylation were marked in green. B. The amino acid sequences of predicted serotonin receptors ortologues with best scores in blast searches with the cestode Eca-5-HT$_{1a}$ were aligned using the ClustalW method. The new 5-HT1 type cloned receptors' names are marked in bold and aligned against sequences of orthologous receptors from *Hydatigera taeniaeformis* (Hta-5-HT$_{1a}$, gene model number TTAC_0000125301), *Taenia asiatica* (Tas-5-HT$_{1a}$, gene model number TASK_0000705401), *Hymenolepis nana* (Hna-5-HT$_{1a}$, gene model number HNAJ_0000723201), *Echinococcus granulosus* (Egr-5-HT$_{1a}$, gene model number EgrG_001050800.1), *Taenia saginata* (Tsa-5-HT$_{1a}$, gene model number TSAs00002g00673m00001), *Echinococcus multilocularis* (Emu-5-HT$_{1a}$, gene model number EmuJ_001050800.1), *Taenia solium* (Tso-5-HT$_{1a}$, gene model number TsM_000928200), *Hymenolepis diminuta* (Hdi-5-HT$_{1a}$, gene model number HDID_0000514801), *Taenia multiceps* (Tmu-5-HT$_{1a}$, gene model number Tm1G003304), *Schistocephalus solidus* (Sso-5-HT$_{1a}$, gene model number SSLN_0001996401), *Dibothriocephalus latus* (Dla-5-HT$_{1a}$, gene model number DILT_0000437701), *Spirometra erinaceieuropaei* (Ser-5-HT$_{1a}$, gene model number SPER_0002466701) and *Gyrodactylus salaris* (Gsa-5-HT$_{1a}$, gene model number scf7180006953168). The transmembrane (TM), intracellular (ICL) and extracellular (ECL) domains are indicated above each alignment. For the sake of simplicity, the amino terminal end, the intracellular loop three and the carboxy terminal end were trimmed partially or completely. The position of residues involved in G-protein coupling are indicated with asterisks below each alignment. Residues present in the new predicted receptors that were not seen in other GPCRs are underlined. Critical residues involved in ligand binding and receptor

function were indicated in bold below each alignment. Cysteine residues potentially involved in disulphide bond formation are marked as S-S between cysteines. The reader is referred to the S1 Table for the complete list of receptor names, species and identification numbers. C. The amino acid sequences of predicted cestode serotonin receptors were aligned with a repertoire of serotonin receptors cloned from *Echinococcus canadensis* (Eca-5-HT$_{1b}$, gene model EcG7_00799; Eca-5-HT$_{7a}$ and Eca-5-HT$_{7b}$, Genbank accession numbers MH707372 and MH707373 respectively), *Mesocestoides vogae* (Mvo-5-HT$_{7a}$, Genbank accession numer MH707374), *Dugesia japonica* (Dja-5-HT$_{1a}$ to Dja-5-HT$_{1e}$, Dja-5-HT$_{2a}$ to Dja-5-HT$_{2c}$ and Dja-5-HT$_{7a}$ to Dja-5-HT$_{7h}$; PMID PMC4569474), *Schistosoma mansoni* (Sma-5-HT$_1$ and Sma-5-HT$_{7b}$, Genbank accession numbers XP_018645423 and KX150867 respectively), *Caenorhabditis elegans* (Cel-5-HT$_1$, Cel-5-HT$_2$ and Cel-5-HT$_7$, Uniprot accession numbers G5EGH0, O17470 and Q22895 respectively) and *Drosophila melanogaster* (Dme-5-HT$_{1a}$, Dme-5-HT$_{1b}$, Dme-5-HT$_{2a}$, Dme-5-HT$_{2b}$ and Dme-5-HT$_7$, Genbank accession numbers CAA77570.1, CAA77571.1, CAA57429.1, NP_001262373.1 and NP_524599.1 respectively). The alignment included representative examples of the three major classes of serotonergic GPCRs from invertebrates. These include type 2 serotonin receptors (5-HT2, blue), type 7 (5-HT7, green) and type 1 receptors (5-HT1, red). The cestode GPCR sequences cloned and functionally expressed in this study (name labels shaded in gray: Eca-5-HT$_{1a}$, Mvo-5-HT$_{1a}$ and Hmi-5-HT$_{1a}$) as well as the putative receptor identified in databases (name label shaded in gray: Eca-5-HT$_{1b}$) cluster within a clade of 5-HT1 like receptors. The FMRFaR neuropeptide receptor from *D. melanogaster* (DmeFMRFaR, Uniprot accession number Q9VZW5) was used as outgroup. The tree was tested by bootstrap analysis with 500 iterations. The length of the branches is proportional to the genetic distance between sequences (see scale bar).

receptor is usually characterized by Gα$_{i/o}$ coupling upon ligand activation and an inhibitory effect on cAMP production.

We have also retrieved another sequence (EcG7_00799) from the *E. canadensis* genome belonging to the 5-HT1 type christened Eca-5-HT$_{1b}$ (S3 Fig). The nucleotide sequence of the cDNA for this gen was 1638 bp long with an open reading frame of 1638 bp. The gene has three coding exons separated by two introns. Similar to Eca-5-HT$_{1a}$, multiple sequence alignments with other orthologous sequences suggested that Eca-5-HT$_{1b}$ is also complete. The predicted amino acid sequence is 545 residues long with a deduced molecular weight of 61.2 kDa. The prediction of N-linked glycosylation site, marked one potential site of N-linked glycosylation at the N-terminal end (S3A Fig). The hypothetical Eca-5-HT$_{1b}$ receptor was also conserved in other cestode species (S3B Fig). In a similar way to Eca-5-HT$_{1a}$, Eca-5-HT$_{1b}$ grouped within the 5-HT1 clade (Fig 1C). The predicted topology of this cestode receptor comprised the canonical fold of seven transmembrane domains (7TM), with an extracellular amino terminal end, an intracellular carboxy terminal end and three alternating ICLs and ECLs (S3A Fig). However, this annotated receptor was not examined experimentally.

For labelling important amino acid residues, the Ballesteros and Weinstein nomenclature for amino acid residue numbering was followed [25]. Most of the residues identified as having major roles in mammalian GPCR function are conserved in invertebrate sequences (Table 1). Residues involved in ligand binding in mammalian serotonin GPCRs were conserved, including aspartate residue D$^{3.32}$ (the carboxylate group of which stabilizes the ligand in the orthosteric pocket) and V$^{3.33}$, C$^{3.36}$, W$^{6.48}$ and F$^{6.51}$ (the side chains of which form a narrow hydrophobic cleft important for ligand recognition). Moreover, the indole N-H hydrogen of 5-HT forms a hydrogen bond with the threonine residue T$^{3.37}$, which is also conserved. Examples of residues conserved in other GPCRs but not exclusively in serotonergic GPCRs are the cysteine of the third transmembrane domain and the cysteine present in the ECL 2 that partially covers the ligand binding pocket (C$^{3.25}$-C$^{ECL2}$) [21]. The highly conserved NPxxY motif toward the cytoplasmic end of TM7 involved in GPCR activation [44] is also conserved in the cestode GPCR sequences (Fig 1B).

Table 1 summarizes critical amino acid residues at the indicated positions (B&W nomenclature) in the human Hsa-5-HT$_{1b}$ receptor. Corresponding residues in invertebrate GPCR sequences are shown, including the GPCRs cloned here (marked in bold in the left), 5-HT1, 5-HT2 and 5-HT7 receptors from *Dugesia japonica* (Dja-5-HT), 5-HT1 and 5-HT7 receptors from *Schistosoma mansoni* (Sma-5-HT), 5-HT1 and 5-HT7 receptors from *Echinococcus canadensis* (Eca-5-HT) and finally, the 5-HT7 receptor from *Mesocestoides vogae* (Mvo-5-HT$_{7a}$).

The presence of certain residues at defined positions in cestode GPCR sequences (Table 1) suggests that the identified sequences belong to the 5-HT1 class. For example, in invertebrates

**Table 1. Important amino acid residues in ligand binding and receptor activation of the cloned cestode GPCRs and homologous invertebrate GPCRs.**

| | TM 3 | | | | | | | ICL 2 | | | ECL 2 | TM 5 | | | TM 6 | | | | | |
|---|---|---|---|---|---|---|---|---|---|---|---|---|---|---|---|---|---|---|---|---|
| | F$^{3.28}$ | I$^{3.29}$ | D$^{3.32}$ | V$^{3.33}$ | C$^{3.36}$ | T$^{3.37}$ | I$^{3.40}$ | D$^{3.49}$ | R$^{3.50}$ | Y$^{3.51}$ | C$^{ECL2}$ | T$^{5.39}$ | S$^{5.42}$ | A$^{5.46}$ | F$^{6.44}$ | W$^{6.48}$ | F$^{6.51}$ | F$^{6.52}$ | A$^{6.55}$ | L$^{6.57}$ |
| Hsa-5-HT$_{1b}$ | | | | | | | | | | | | | | | | | | | | |
| Eca-5-HT$_{1a}$ | W | I | D | V | C | T | T | E | R | C | C | **T** | **S** | A | F | W | F | N | A | L |
| Mvo-5-HT$_{1a}$ | W | I | D | V | C | T | T | E | R | C | C | **T** | **S** | A | F | W | F | N | A | L |
| Hmi-5-HT$_{1a}$ | W | I | D | V | C | T | T | E | R | A | C | **T** | **S** | A | F | W | F | N | A | L |
| Eca-5-HT$_{1b}$ | W | I | D | V | C | T | I | D | R | Y | C | **T** | **S** | S | F | W | F | S | A | V |
| Dja-5-HT$_{1a}$ | W | V | D | V | C | S | I | D | R | Y | C | **T** | **S** | A | F | W | F | F | A | I |
| Dja-5-HT$_{1c}$ | W | I | D | V | C | T | I | D | R | Y | C | I | A | S | F | W | F | G | N | L |
| Sma-5-HT$_{1}$ | W | I | D | V | C | T | I | D | R | Y | C | **T** | **S** | A | F | W | F | F | T | I |
| Dja-5-HT$_{2b}$ | W | Y | D | V | T | S | I | D | R | Y | G | I | A | T | F | Y | F | F | Y | L |
| Dja-5-HT$_{2c}$ | W | Y | D | V | T | A | I | D | R | Y | G | I | A | T | F | Y | F | F | Y | C |
| Eca-5-HT$_{7a}$ | Y | S | D | V | C | T | I | D | R | Y | C | Q | A | A | F | W | F | F | Q | I |
| Eca-5-HT$_{7b}$ | F | I | D | V | C | T | I | D | R | Y | C | Q | A | A | F | W | F | F | A | G |
| Mvo-5-HT$_{7a}$ | Y | N | D | V | C | T | I | D | R | Y | C | Q | A | A | F | W | F | F | Q | I |
| Dja-5-HT$_{7a}$ | Y | N | D | V | C | T | I | D | R | Y | C | Q | A | A | F | W | F | F | Q | L |
| Dja-5-HT$_{7f}$ | F | I | D | V | C | S | I | D | R | Y | C | Q | A | A | F | W | F | F | Q | I |
| Sma-5-HT$_{7b}$ | Y | I | D | V | C | T | I | D | R | Y | C | Q | A | A | F | W | F | F | Q | L |

Important residues usually present in receptors of the 5-HT1 type but not commonly observed in other receptor clades were marked in bold. Receptors analyzed in this study were shaded in grey. TM indicates the number of transmembrane domain, ICL, the number of Intracellular loop and finally, ECL, the number of extracellular loop in each receptor.

the tryptophan at the aromatic residue position 3.28 occurs only in 5-HT1 and 5-HT2 receptors but not in the 5-HT7 type GPCRs where mostly tyrosine or phenylalanine residues are present. The cysteine and threonine at positions 3.36 and 3.37 respectively seem to be more frequent in the 5-HT1 and 5-HT7 type than in the 5-HT2 type receptors. Residues at positions 5.42 and 5.46, both in the orthosteric binding pocket, were previously defined as diagnostic of the different serotonergic clades for planarians [15]. Fig 1 and Table 1 show the presence of serine/alanine at the positions 5.42 and 5.46 respectively in TM5 in the three identified cestode GPCRs and in other invertebrate 5-HT1 receptors, which also suggest that they belong to the 5-HT1 class. The Eca-5-HT$_{1b}$ receptor harbors a serine residue at position 5.46 in the TM5, forming an orthosteric binding pocket of higher polarity than the other three sequences with alanine at this position. The other 5-HT receptor types show the presence of non-polar residue alanine in residue 5.42. In invertebrate 5-HT2 receptors a pattern of alanine/threonine is observed, while 5-HT7 receptors interestingly show an alanine/alanine pattern, with only non-polar residues. Very close to these positions, the presence of threonine at position 5.39, a residue in the extended binding site related to ligand selectivity of serotonergic receptor types in humans (21), seems to be a feature of receptors belonging to the 5-HT1 type. Interestingly, in the 5-HT7 receptors cloned by us and others, this position has been replaced with a glutamine residue (Table 1).

Differences were also observed between sequences of the cloned cestode GPCRs and previously characterized invertebrate-vertebrate GPCRs. For example, the DRY motif, which is involved in GPCR activation and almost invariably conserved in the vast majority of GPCRs including other invertebrate sequences [44], was replaced by the sequence ERC or ERA in the three cloned cestode sequences but not in the predicted Eca-5-HT$_{1b}$ (Fig 1B and Table 1). In the PIF motif (5.50, 3.40 and 6.44) that is involved in receptor activation [21], the isoleucine (I 3.40) is replaced by a threonine residue in all three cloned cestode sequences (Table 1). In the sixth transmembrane region, of the two phenylalanines usually seen at positions 6.51 and 6.52,

**Table 2. Residues potentially involved in receptor and G-protein coupling.**

| | TM 5 | | | | TM 7 | |
|---|---|---|---|---|---|---|
| **Hsa-5-HT$_{1b}$** | **I$^{5.61}$** | **A$^{5.65}$** | **R$^{5.68}$** | **I$^{5.69}$** | **S$^{7.57}$** | **N$^{7.58}$** |
| **Eca-5-HT$_{1a}$** | I | A | H | **I** | **S** | K |
| **Mvo-5-HT$_{1a}$** | I | A | H | **I** | **S** | K |
| **Hmi-5-HT$_{1a}$** | I | A | H | **I** | **S** | K |
| **Eca-5-HT$_{1b}$** | I | A | R | **I** | **S** | P |
| **Dja-5-HT$_{1a}$** | I | A | R | **I** | **S** | P |
| **Dja-5-HT$_{1c}$** | I | I | Q | T | **S** | T |
| **Sma-5-HT$_1$** | I | A | R | **I** | **S** | P |
| **Dja-5-HT$_{2b}$** | T | I | Q | T | N | P |
| **Dja-5-HT$_{2c}$** | T | I | Q | T | N | P |
| **Eca-5-HT$_{7a}$** | I | A | M | A | N | R |
| **Eca-5-HT$_{7b}$** | I | A | I | V | N | R |
| **Mvo-5-HT$_{7a}$** | I | A | M | A | N | R |
| **Dja-5-HT$_{7a}$** | I | A | M | A | N | R |
| **Dja-5-HT$_{7f}$** | I | T | M | V | N | R |
| **Sma-5-HT$_{7b}$** | I | A | M | S | D | R |

Important residues usually present in receptors of the 5-HT1 type but not commonly observed in other receptor clades were marked in bold. Receptors analyzed in this study were shaded in grey. TM indicates the number of transmembrane domain in each receptor.

involved in ligand binding and likely also in receptor activation (44), 6.52 was replaced by asparagine in the three sequences reported here and by serine in the hypothetical Eca-5-HT$_{1b}$.

Previous work has investigated the structural basis of the interaction between the Hsa-5-HT$_{1b}$ receptor and Gα$_{i/o}$ [22]. We therefore analyzed residues in flatworm GPCRs that are potentially involved in Gα$_{i/o}$ coupling (Table 2 and Fig 1B).

A non-polar isoleucine residue was present at position 5.69 in almost all of the sequences belonging to the 5-HT1 (with the exception of Dja-5-HT$_{1c}$ in which this residue was threonine) whereas threonine, alanine, valine or serine were more frequent in other invertebrate clades. The serine at position 7.57 was only seen in 5-HT1 type receptors, whereas it is replaced by asparagine in almost all GPCRs from the 5-HT2 and 5-HT7 types.

Since the four receptors reported in this work were predicted to belong to the 5-HT1 type, it is expected that these GPCRs couple with the Gα$_{i/o}$ family of G proteins. To predict the coupling specificity of the three serotonergic receptors sequenced, the program PRED-COUPLE 2.00 was used (Table 3). The four cestode receptors have high scores for Gα$_{i/o}$ coupling partners, with normalized scores of 0.52 for Eca-5-HT$_{1a}$, 0.98 for Eca-5-HT$_{1b}$, 0.44 for Mvo-5-HT$_{1a}$ and 0.60 for Hmi-5-HT$_{1a}$. However, for the Hmi-5-HT$_{1a}$, a higher normalized score of 0.93 for the Gα$_{q/11}$ coupling partner was observed, suggesting that this receptor could have some degree of promiscuity at the type of G protein interaction, increasing the possibility of interaction with both types of G proteins: Gα$_{q/11}$ and Gα$_{i/o}$. Therefore, the results shown here seems to arrive to similar conclusions obtained by phylogenetic analyses.

Next, homology modeling was performed to determine important structural features of the three cloned GPCRs. For these receptors, Ramachadran plots of the modeled proteins showed that 95.7% or more of all residues were in favored regions, while 99.3% or more residues were in allowed regions and only 0.7% or less residues were in outlier regions of the Ramachandran plots (Fig A–H in S2 Text). In addition, these structure homology models obtained showed ERRAT values higher than 92.9263 and QMEN values close to -4.0, indicating a good quality of the homology structure models. All models showed RMSD values lower than 0.73 Å and

**Table 3. Prediction of coupling specificity for the 5-HT1 type of cestode receptors.**

| G-Protein coupled receptor (GPCR) | Gα-protein coupling specificity—normalized score (PRED-COUPLE 2.00) |
|---|---|
| **Eca-5-HT$_{1a}$** | Gα$_{i/o}$ - **0.52** |
| | Gα$_s$ - 0.27 |
| | Gα$_{q/11}$ - 0.02 |
| | Gα$_{12/13}$ - 0.00 |
| **Mvo-5-HT$_{1a}$** | Gα$_{i/o}$ - **0.44** |
| | Gα$_s$ - 0.17 |
| | Gα$_{q/11}$ - 0.08 |
| | Gα$_{12/13}$ - 0.00 |
| **Hmi-5-HT$_{1a}$** | Gα$_{q/11}$ - **0.93** |
| | Gα$_{i/o}$ - **0.60** |
| | Gα$_s$ - 0.00 |
| | Gα$_{12/13}$ - 0.00 |
| **Eca-5-HT$_{1b}$** | Gα$_{i/o}$—**0.98** |
| | Gα$_{q/11}$—0.01 |
| | Gα$_s$—0.00 |
| | Gα$_{12/13}$—0.00 |

Scores were obtained with the program PRED-COUPLE 2.0 which reflects the probability of coupling (from 0, no coupling; to 1, coupling) of the three receptors sequenced in this work with each type of G protein. The best scores according to the program were marked in bold.

TM-align values higher than 0.92725, suggesting that the cestode 5-HT1 receptors studied in this work have a similar structure as Hsa-5-HT$_{1b}$ and a good superposition between these proteins (Table 1 in S2 Text). Homology modeling of the three cestode sequences in this work compared with the human 5-HT$_{1b}$ receptor (Fig 2), revealed high structural similarity at positions and orientations of critical residues like 3.32; 3.37; 5.42 and 5.46 which form part of the orthosteric binding pocket in 5-HT1 receptors (Fig 2B–2E).

Additional homology modeling compared the structure of the cloned cestode 5-HT1 GPCRs with 5-HT7 receptors previously reported from cestodes [10], (Fig 2F–2Q and Table 1 in S2 Text). Comparative analysis of the residues involved in receptor binding and G protein coupling showed that both types of receptors (5-HT1 vs 5-HT7) have important structural differences (Fig 2). Critical residues implicated as 5-HT1 type specific (shown in Table 1), for example at positions 5.39 (threonine 5-HT1, glutamine 5-HT7) and 5.42 (polar serine in 5-HT1, nonpolar alanine in 5-HT7) are indeed located in the orthosteric binding pocket of the cestode GPCRs (Fig 2F, 2G, 2J, 2K, 2N & 2O). The tridimensional structure in the 5-HT1 type suggests that the threonine (5.39) is very close to the critical polar serine residue at position 5.42 and points to a higher polarity of the 5-HT1 binding pocket with respect to 5-HT7. Regarding the putative G protein coupling residues, the models reveal that TM5 residues (isoleucine 5.61, alanine 5.65, isoleucine 5.69 and histidine 5.68) and TM7 residues (serine 7.57 and lysine 7.58) are located in positions likely accessible to the G protein from the cytoplasmic side. The coupling residues differ depending of the GPCR type as shown in Table 2. The isoleucine at position 5.69 in 5-HT1 type is replaced by the shorter alanine in the Eca-5-HT$_{7a}$ or valine in Eca-5-HT$_{7b}$ receptors respectively. The hydroxyl group of serine at position 7.57 in 5-HT1 is replaced by the bulkier amide group from asparagine in almost all of the 5-HT7 receptors analyzed (Fig 2H, 2I, 2L, 2M, 2P & 2Q). The inclusion of the nitrogen atom here could be a selectivity determinant for the interaction with the alpha subunit of the G-protein at this position.

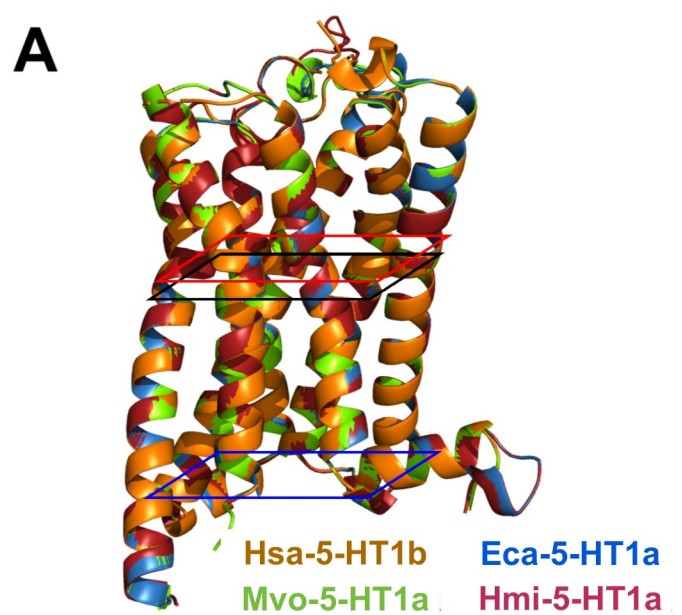

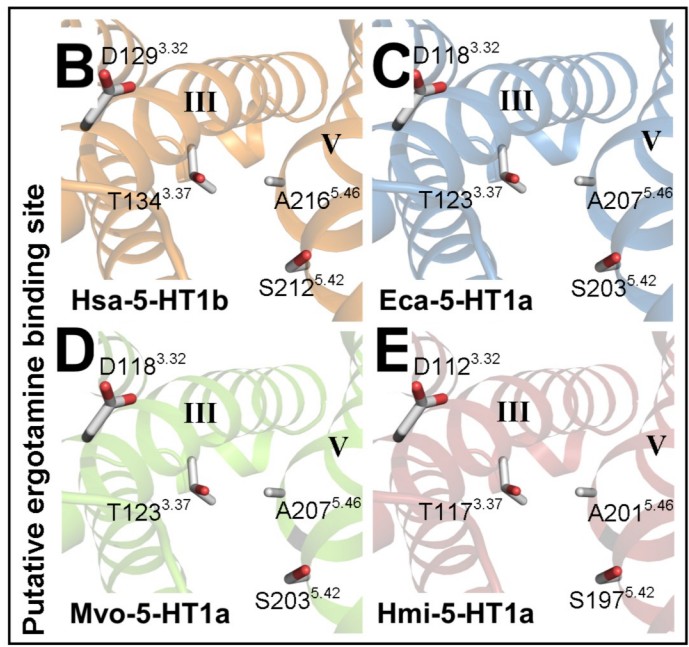

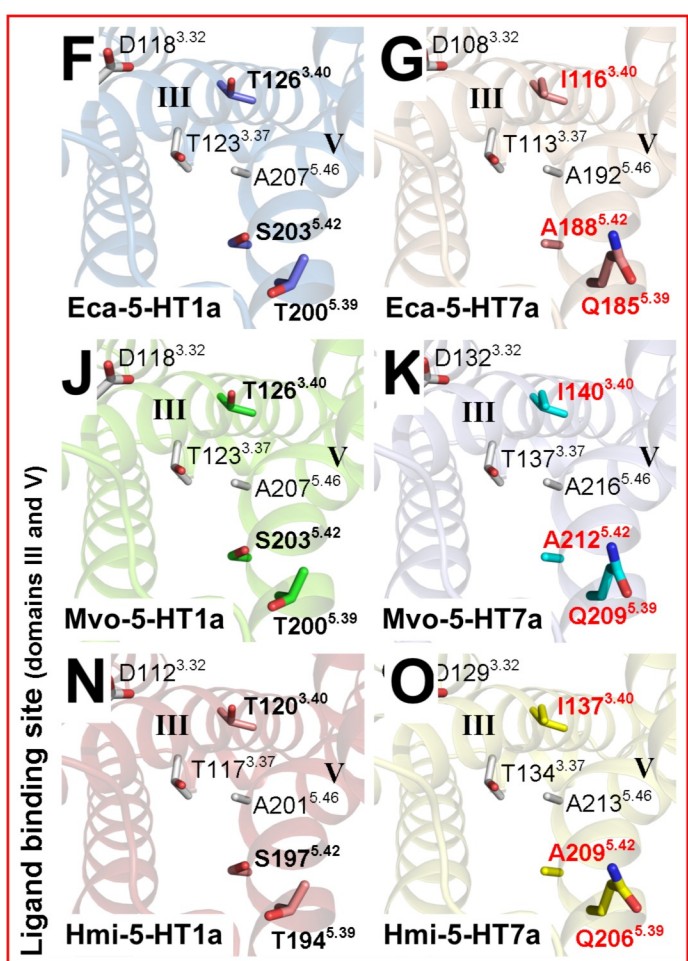

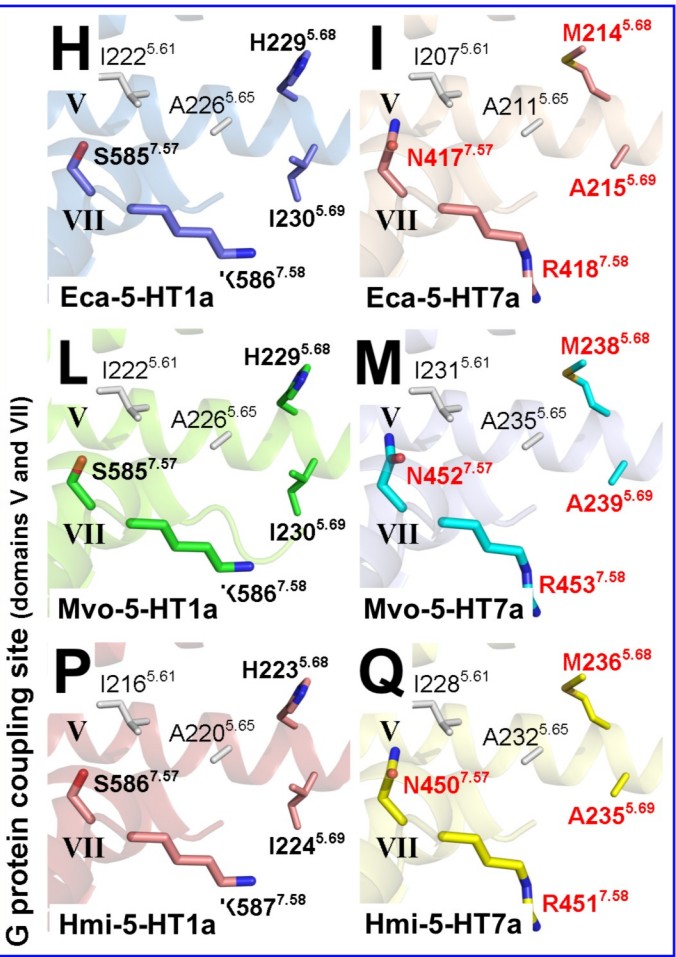

**Fig 2. Structural comparative analysis of cestode 5-HT1 type serotonergic receptors.** (A-E) Structural similarities and differences among the 5-HT1 type serotonergic receptors of *Homo sapiens* (Hsa-5-HT$_{1b}$) and cestodes. (A) Superposition of crystallized Hsa-5-HT$_{1b}$ (orange; PDB 4IAR) and the homology models of Eca-5-HT$_{1a}$ (blue; *Echinococcus canadensis* G7), Mvo-5-HT$_{1a}$ (green; *Mesocestoides vogae*), and Hmi-5-HT$_{1a}$ (red; *Hymenolepis microstoma*). (B-E) Close-up views of the residues involved in the putative ergotamine interaction site of (B) Hsa-5-HT$_{1b}$, (C) Eca-5-HT$_{1a}$, (D), Mvo-5-HT$_{1a}$, and (E) Hmi-5-HT$_{1a}$, in the transmembrane domains III (labeled III) and V (labeled V). (F-Q) Comparative analysis of residues involved in the ligand binding and G protein coupling sites between cestode 5-HT1 and 5-HT7 type serotonergic receptors. (F, G, J, K, N and O) Close-up views of residues involved in the ligand binding site of (F) Eca-5-HT$_{1a}$, (G) Eca-5-HT$_{7a}$, (J), Mvo-5-HT$_{1a}$, (K), Mvo-5-HT$_{7a}$, (N), Hmi-5-HT$_{1a}$, and (O) Hmi-5-HT$_{7a}$, in the transmembrane domains III (labeled III) and V (labeled V). Note that threonines at positions 3.40 and 5.39 and serine at 5.42 (in bold) in 5-HT1a receptors are replaced by isoleucine, glutamine, and alanine (in red); respectively, in 5-HT7a receptors. (H, I, L, M, P and Q) Close-up views of residues involved in the G protein coupling site of (H) Eca-5-HT$_{1a}$, (I) Eca-5-HT$_{7a}$, (L), Mvo-5-HT$_{1a}$, (M), Mvo-5-HT$_{7a}$, (P), Hmi-5-HT$_{1a}$, and (Q) Hmi-5-HT$_{7a}$, in the transmembrane domains V (labeled V) and VII (labeled VII). Note that histidine, isoleucine, serine, and lysine at the positions 5.68, 5.69, 7.57, and 7.58 (in bold); respectively, in 5-HT1a receptors are replaced by methionine, alanine, asparagines, and arginine residues (in red); respectively, in 5-HT7a receptors. In all the plots, protein structures were represented as cartoons and the residue side chains involved in the putative ergotamine interaction, ligand binding, and G protein coupling sites were represented as sticks.

One important structural characteristic of 5-HT1 receptors is the presence of a long intra-cellular loop 3 and the small carboxy terminal end (S4A Fig). By contrast, serotonergic receptors from the clade 7 are characterized by smaller third intracellular loop 3 and bigger carboxy terminal end (S4B Fig). Similar findings are obtained when the human 5-HT1 and 5-HT7 are compared between them (S4C versus S4D Fig).

## Functional characterization of Eca-5-HT$_{1a}$

The cDNA encoding Eca-5-HT$_{1a}$ was transiently transfected into the stable HEK293-GNA15 cell line. In this line, the gene GNA15 encodes the promiscuous G$\alpha_{15}$ protein that binds to GPCRs irrespective of their native coupling type and activates calcium release through the G$\alpha_q$ pathway [45]. The dataset here obtained shows that the addition of serotonin to cells transfected with cDNA for Eca-5-HT$_{1a}$ induced calcium release from intracellular stores in a concentration dependent manner (Fig 3A).

The EC50 for serotonin was 7.74 ± 0.63 nM, which shows that the receptor has a high affinity for serotonin. This response was not seen when the cell line was not transfected (Fig 3B). In order to determine if the observed response was specific for serotonin, the transfected GNA15-cells were exposed to other ligands at the same concentration than serotonin (10 μM). The results clearly showed that Eca-5-HT$_{1a}$ responded to serotonin and also, but to a lesser degree, to tryptamine but not to other ligands (biogenic amines or neurotransmitters) confirming the predictions suggested by bioinformatic analysis (Fig 3C).

As it was showed previously in other species, especially in humans, residues located at positions 3.32, 3.36, 3.37 and 6.48 besides others, are playing central roles in receptor binding and function [21]. In order to see if these residues could play also essential roles in cestode receptor function, three residues belonging to the putative ligand binding pocket and one residue conforming the molecular toggle switch, were mutated to alanine in Eca-5-HT$_{1a}$. As it can be seen in the Fig 3D, the four mutants shown severe impact in receptor function. It was observed that mutants with substitutions at positions 3.32 and 6.48 resulted in complete lack of activity. Substitutions at positions 3.36 and 3.37 resulted in a severe impaired (but mensurable) activity in both mutants, with EC50s of 14.6 and 11.5 μM respectively. The Fig 3E shows the molecular model of the orthosteric binding site in the Eca-5-HT$_{1a}$ receptor. The localization of the mutated residues in the serotonin binding site suggest major roles for such residues in ligand binding and receptor function.

## Pharmacological characterization of Eca-5-HT$_{1a}$

In order to pharmacologically characterize Eca-5-HT$_{1a}$, we used HEK293 cells stably expressing the promiscuous G$\alpha$15 subunit and the Eca-5-HT$_{1a}$ receptor (HEK293-GNA15-Eca-

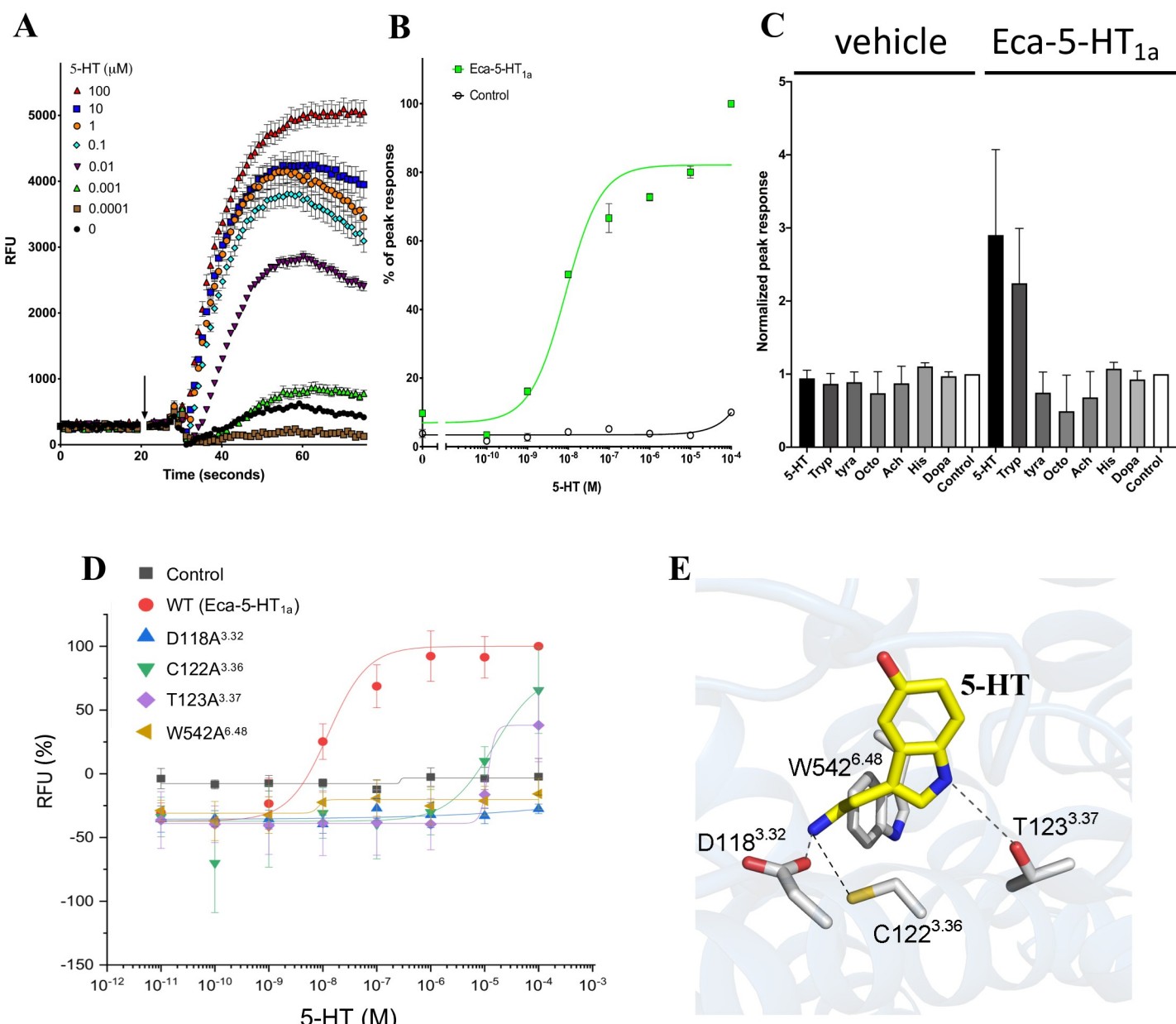

**Fig 3. Heterologous expression of Eca-5-HT$_{1a}$ receptor.** A) Time resolved measurements of Ca$^{2+}$ accumulation (measured as raw fluorescence units, RFU) in cells expressing the cestode Eca-5-HT$_{1a}$ before and after addition of different concentrations of 5-HT (arrow, concentrations indicated in legend in μM). B) Concentration-response curve measuring peak amplitude of 5-HT evoked fluorescence change (measured as a percentage) in cells expressing Eca-5-HT$_{1a}$ (green closed squares), or untransfected HEK293-GNA15 cell line (open circles). C) Bar graph showing normalized peak responses in cells transfected with the new receptor to indicated neurotransmitters (10 μM). D) Concentration-response curve measuring 5-HT evoked fluorescence change (measured as raw fluorescence units, RFU) in cells expressing wild type Eca-5-HT$_{1a}$ (red solid dot), or the following mutants: D118A$^{3.32}$ (blue triangle), C122A$^{3.36}$ (green inverted triangle), T123A$^{3.37}$ (purple rhombus) and W542A$^{6.48}$ (brown triangle). Black solid squares represent the untransfected control (HEK293-GNA15 cell line). E) molecular model of the orthosteric binding pocket in Eca-5-HT$_{1a}$, showing the role of the mutated residues in serotonin binding. The molecule shown in yellow represent 5-HT. The gray sticks represent the R groups from the amino acids mutated here. In all cases, the amine groups are shown in blue, thiol group is shown in yellow and hydroxyl groups in red. Hypothetical interactions between the 5-HT molecule and receptor mutated residues were marked by dotted lines. Abbreviations are: 5-HT, serotonin; Tyra, tyramine; Octo, octopamine; Ach, acetylcholine; His, histidine; Dopa, dopamine. For panels A) and B), each point represents the average of three biological experiments with three technical replicates each. For panel C), each point represents the average of two biological experiments with three technical replicates each. Error bars corresponds to the standard error of the mean.

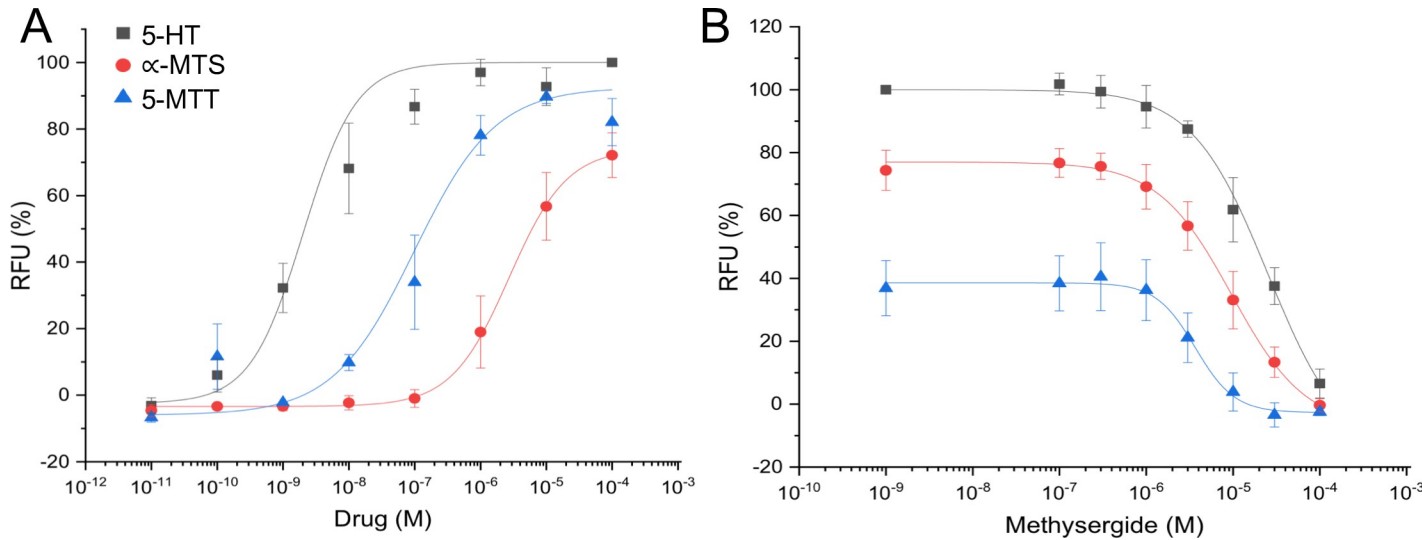

**Fig 4. Effect of various agonists and antagonist on Eca-5-HT$_{1a}$ in HEK293-GNA15 cells.** (A) Concentration-dependent activation of Eca-5-HT$_{1a}$ with 5-HT (black box) or the 5-HT receptor agonists α-methylserotonin (α-MTS, red ball) and 5-methoxytryptamine (5-MTT, blue triangle), shown as the percentage of activation achieved with 5-HT (maximum response = 100%). (B) Concentration-dependent inhibition of Eca-5-HT$_{1a}$ obtained with methysergide on 5-HT (0.03 μM), α-methylserotonin (10 μM) and 5-methoxytryptamine (1 μM) mediated activation. In all cases, data represent the mean of three independent measurements (each performed in triplicate).

5-HT1a stable cell line). No 5-HT evoked signal was observed in non-transfected cells (HEK293-GNA15 stable cells, results not shown). Agonist responses were obtained when cells expressing Eca-5-HT$_{1a}$ were incubated with increasing 5-HT or the 5-HT receptor agonists α-methylserotonin and 5-methoxytryptamine (Fig 4A). The concentration-response relationship for 5-HT and these agonists was examined at concentrations ranging from 10 pM to 100 μM (Fig 4A). The resulting sigmoidal concentration-response curve of 5-HT shows receptor activation in a concentration-dependent and saturable manner. Half-maximal activation (EC50) was achieved at 5-HT concentrations of 0.002 ± 0.001 μM. The response to 5-HT was followed by 5-methoxytryptamine (EC50 = 0.098 ± 0.05 μM) and finally, by α-methylserotonin (EC50 = 2.73 ± 0.73 μM). By the contrary, methysergide, a non-selective ligand of mammalian serotonin receptors exhibited a concentration-dependent curve of inhibition (Fig 4B).

## Eca-5-HT$_{1a}$ receptor showed a strong pattern of staining typical of the nervous system and suggests a major role in the neuronal function

Whole mount immunofluorescence assays in protoscoleces of *E. canadensis*, using an antibody against Eca-5-HT$_{1a}$, revealed a pattern typical of the nervous system with two important masses of stained nerves forming the bilobed brain in the scolex region from which two longitudinal nerve cords projected from the anterior to the posterior end (Fig 5B and 5D).

In some pictures, nerve commissures were seen connecting the two longitudinal nerve cords (Fig 5D) and the two lobes of the brain (Fig 5H). When this pattern is compared to the pattern of staining with an anti-tropomyosin antibody (a muscle stain) [44], it can be clearly observed that Eca-5-HT$_{1a}$ does not colocalize with the muscular system (Fig 5B and 5D).

With the aim to elucidate the relationship between this receptor and the neurotransmitter serotonin, colocalization studies were performed using mouse antibodies against the cestode GPCR marked with rhodamine (red) and rabbit antibodies against serotonin marked with FITC (green). The pictures suggest that there is not complete colocalization between serotonin and Eca-5-HT$_{1a}$ signals (Fig 5I, 5J and 5N). In the scolex region, the receptor seems to be in

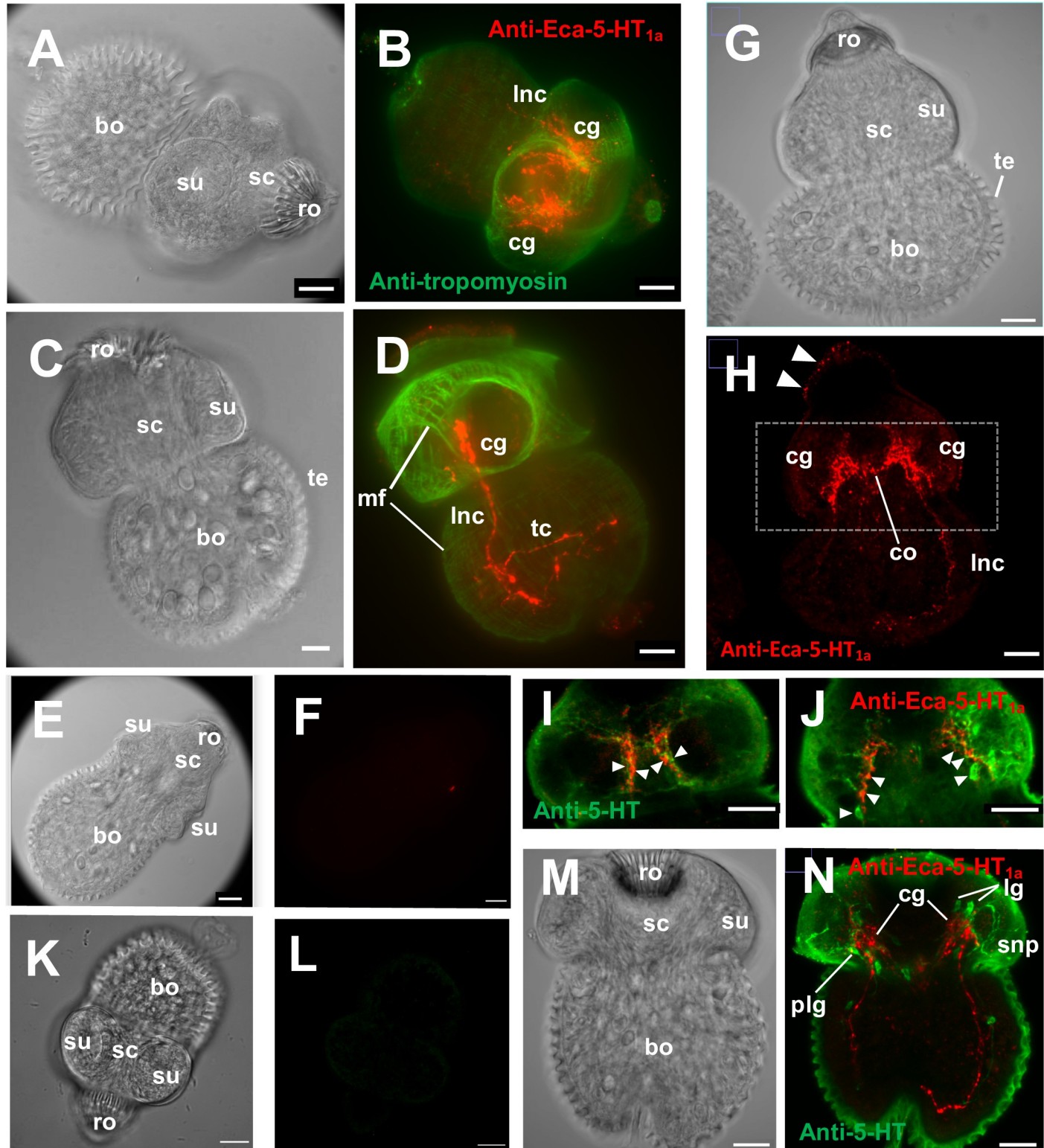

**Fig 5. Inmunolocalization of Eca-5-HT₁ₐ in protoscoleces of *Echinococcus canadensis*.** Protoscoleces were probed with anti-Eca-5-HT₁ₐ antibody (red) and anti-tropomyosin antibody or anti-serotonin antibody (green) and visualized by confocal microscopy. A) Phase contrast view of the protoscolex shown in B. B) Fluorescent image of the same protoscolex co-labelled with anti-Eca-5-HT₁ₐ antibody and anti-tropomyosin antibody. No evidence of proximity between the two signals were found. Intense signal was found in the cerebral ganglia (cg) and the main longitudinal nerve cords (lnc). C) Phase contrast view of the protoscolex shown in D. D) Fluorescent image of protoscolex co-labelled with anti-Eca-5-HT₁ₐ antibody and anti-tropomyosin antibody. The intense green signal shows the complex

pattern of muscle fibers (mf). The strong red signal was observed in the cerebral ganglia (cg), the longitudinal nerve cord (lnc) and transverse commissure (tc). E) Phase contrast of the protoscolex shown in F. F) Fluorescent image of protoscolex labelled with preimmune serum obtained from the same mice which were later inoculated with Eca-5-HT$_{1a}$. G) Phase contrast view of the protoscolex shown in H. H) Fluorescent image of the same protoscolex labelled with anti-Eca-5-HT$_{1a}$ antibody (red). Strong Eca-5-HT$_{1a}$ signal was found in the cerebral ganglia (cg) and the main longitudinal nerve cords (lnc). Some weak signal (arrowheads) could be seen in the surface region of the scolex. I) and J) Two different stack integrations and picture magnifications of the fluorescent image of the protoscolex shown in H. Alternating regions of receptor and serotonin immunoreactivity can be seen (arrowheads) in the scolex region. K) Phase contrast of the protoscolex shown in L. L) Fluorescent image of protoscolex incubated without primary antibody. M) Phase contrast view of the protoscolex shown in N. N) Fluorescent image of the same protoscolex co-labelled with anti-Eca-5-HT$_{1a}$ antibody and anti-serotonin antibody. Strong signal was observed in the cerebral ganglia (cg), the lateral nerve cord (lnc) and the cerebral commissure (co). For the phase contrast images, the tegument (te), body (bo), scolex region (sc) with sucker (su) and the rostellum (ro) were marked. Scale bar 20 μm.

nervous fibers that are contiguous and intertwined but not exactly coincident with the fibers obtained with the anti-serotonin antibody (Fig 5I and 5J). Some pictures also show faint receptor signals in the surface of the scolex region of the worm (Fig 5H). In the body region, some points of contact can be seen between green (serotonergic) and receptor (red) fibers but the overall impression is that there are less contiguous regions than in the scolex (Fig 5N). Cell bodies in the major ganglia clearly stained with serotonin antibodies (i.e. lateral, posterior lateral or rostellar ganglia) were not stained with the anti-receptor antibody (Fig 5N). The pattern of staining suggests that receptor-containing fibers could receive serotonergic input from proximal contiguous fibers.

## Discussion

This work reports the initial molecular and functional characterization of a new serotonergic receptor in platyhelminths that likely belongs to the 5-HT1 type of GPCRs. To the best of our knowledge, this is the first functional characterization of a presumptive 5-HT1 type receptor in platyhelminths. Highly conserved orthologues are found in a wide variety of species of cestodes suggesting these receptors could play major roles in the biology of these parasites. The evidence for the presumptive assignment the proteins as a 5-HT1 type of receptors are: 1) phylogenetic analysis, 2) presence of residues in the ligand-binding pocket that are characteristic of 5-HT1 type, 3) presence of putative G-protein coupling residues (selectivity determinants of the receptor) that are more similar to the 5-HT1 type than to other clades, and finally, 4) structural features typical of 5-HT1 type of receptors.

5-HT1 type GPCRs is expected to be coupled to the G$\alpha_{i/o}$ family of G proteins to cause inhibition of cAMP dependent signaling. This represents the opposite outcome to the rise of cAMP after the activation of 5-HT7 type of cestode receptors which were recently deorphanized [10]. Relatively few helminth GPCRs have been deorphanized to date, and further efforts are needed to characterize how they operate and their *in vivo* roles.

In spite of the overall similarity of the novel cestode receptors to the 5-HT1 type receptors, some uniqueness in residues and motifs critical for receptor function were identified: first, the receptor has an ERC or ERA motif after the third transmembrane domain but not the classical DRY domain. Second, the PIF domain is replaced by a PTF domain. Third, phenylalanine at position 6.52, which is an essential component of a cluster of aromatic residues that surrounds tryptophan 6.48, referred as a "receptor toggle switch", is replaced by asparagine [46]. The substituted residues form part of conserved motifs well known as "molecular microswitches" or highly conserved motifs related to the stabilization of GPCRs in activated or inactivated states [44, 47]. The significance of these substitutions is unclear but could be of importance as they suggest potential differences in the mechanisms of receptor activation/inactivation under ligand binding or pharmacology with the human counterparts. These motifs are not only different with other invertebrate 5-HT1 members but also with the mammalian Hsa-5-HT$_{1b}$. Possibly, these differences could be exploited for selective druggability. The hypothetical Eca-

5-HT$_{1b}$ seems to be more conserved at these canonical motifs, however, the substitution of alanine by serine at the key position 5.46, could result in relevant differences in pharmacological profiles with other orthologous receptors from other species.

The heterologous expression studies of the *E. canadensis* receptor confirm that the molecule is functional, responding specifically to the addition of 5-HT in a concentration-dependent and high affinity manner. Tryptamine, a well-known agonist of 5-HT GPCRs [48], also elicited a strong response in the presence of the cestode receptor but to a lesser degree than to 5-HT. Other invertebrate 5-HT1 receptors, for example Trica5-HT1 receptor from the red flour beetle, *Tribolium castaneum* [49], was shown to be insensitive to tryptamine and this could suggest that the cestode receptor could be less selective to other biogenic amines than the insect receptor. The synthetic agonists 5-methoxytryptamine and α-methylserotonin give good responses and both were tested in the previously mentioned Trica5-HT1 with similar responses [49]. However, the sensitivity to 5-methoxytryptamine was higher than to α-methylserotonin and this difference in agonist activity could be explained probably by the lower tolerance to replacements near the primary amine of 5-HT (which seems to make contacts with the aspartate 3.32 and cysteine 3.36 of Eca-5-HT$_{1a}$) than in the proximity of the hydroxyl group. The well-known 5-HT GPCR ligand methysergide, has been described as having agonist and antagonist activities in invertebrates [13]. In this work, good blocking activity was observed with this ligand in Eca-5-HT$_{1a}$. Mutagenesis studies confirms major roles for residues at positions 3.32, 3.36, 3.37 and 6.48, indicating that they could be part of the binding pocket and/or receptor activation mechanism also in cestodes. In mammals, it was postulated that aspartate in position 3.32 is forming a salt bridge with the positively charged amino group of 5-HT [21]. This interaction could be seriously affected when aspartate is replaced by alanine, interfering with ligand binding. Another important residue analyzed was threonine 3.37, which was postulated to form a hydrogen bond with the indole N-H [21]. This type of interaction could be abolished when the polar threonine is replaced by the non-polar alanine in the cestode mutant receptor. However, some activity could be detected using this latter mutant suggesting that this residue is very important but not critical for receptor function. Tryptophan 6.48 is an essential component of aromatic residues referred to as a "toggle switch", which includes phenylalanine 6.44, phenylalanine 6.51 and phenylalanine 6.52 [46]. It is thought that the movement of the tryptophan 6.48 is one of the major features of receptor activation [46]. The results obtained here shown a complete lack of activity after the substitution of tryptophan by alanine at the residue 6.48 suggesting that this replacement could interfere with this mechanism of receptor activation also in cestodes. For residue 3.36, it was postulated that it could stabilize tryptophan 6.48 by Van der Waals interactions. It was postulated that one potential role for residue 3.36 is to stabilize the conformation of the receptor in an inactive state. The replacement of cysteine by alanine could abolish this interaction resulting in a very low activity in the cestode mutant. In summary, bioinformatic analyses predicts residues with major function in Cestode receptors and some of this residues were confirmed to be of paramount importance in receptor function by mutant analyses.

The confocal microscopy performed in protoscoleces of *E. canadensis* revealed a pattern of staining consistent with expression in the central and peripheral nervous system. Some nervous fibers and ganglia were revealed with the antibody against the Eca-5-HT$_{1a}$ receptor but co-localization with an anti-serotonin antibody suggests that serotonin fibers were not the same as those stained with the anti-Eca-5-HT$_{1a}$ antibody. In the images shown, both types of fibers seem to be intertwined like a mesh suggesting that the serotonergic fibers are close enough to release serotonin that could be received by the fibers expressing the GPCR. However, comparisons between receptor and serotonin locations should be interpreted with caution since quiet a lot serotonin background staining could be seen, especially in the tegument

of the worm. In the light of the deep knowledge of serotonin neuroanatomy in *Echinococcus* spp. [14, 41, 50, 51], high levels of serotonergic staining observed in the tegument seems to be an artifact probably due to the low titers and high concentrations of the serotonin antibody used. Anyways, the fluorescence images obtained with the anti-Eca-5-HT$_{1a}$ antibody showed the classical orthogonal pattern with two bilobed structures resembling the brain and two longitudinal cords emerging from this primitive brain. Minor Eca-5-HT$_{1a}$ staining at the tegument surface could also be observed, which suggests the existence of potential sensitive nerve endings [52]. The participation of the receptor in the neuromuscular plaque is not clear: in some pictures some fibers emanated from the central nervous system toward the more superficial muscular system but the images are not conclusive in this way. Similar results of nervous system staining were obtained by Patocka and colleagues [16] for Sm-5-HTR: a 5-HT7 type of GPCR from *S. mansoni*. Using a purified antibody against this receptor the authors also found strong immunoreactive nerve fibers in schistosomulae and adults. An important difference between Eca-5-HT$_{1a}$ and Sm-5-HTR was that, in the case of cestode receptor, no expression could be seen outside from the nervous system whereas Sm-5-HTR was expressed in the digestive system of *S. mansoni* worms. While classical neurotransmitters such as GABA and glutamate appear to function mainly locally at synapses, serotonin can act locally but also in a paracrine way, diffusing several microns from its release sites at concentrations sufficient to activate its receptors [53]. Furthermore, serotonin receptors are often localized at nonsynaptic sites [54]. These observations suggest that serotonin might act as an extrasynaptic signal to activate several receptor types on cells distant from its release [53]. In *C. elegans*, one of the several behaviors controlled by serotonin is enhanced slowing which is characterized by significant lower levels of worm motility after serotonin addition. This response was found to be mediated by SER-4 (Cel-5-HT$_1$) receptor [55]. Interestingly, it was found here that ser-4, represented in the phylogenetic three as Cel-5-HT$_{1a}$ grouped in the same clade than Eca-5-HT$_{1a}$.

In parasitic nematodes, a piperazine derivative and known 5-HT1 agonist was proposed as a potential anthelmintic agent [20]. The evidence presented suggests that compounds with affinity for this type of GPCR could be used as an antiparasitic. On the other hand, the transcriptomic data of the present study suggest a restricted expression to the protoscolex stage. The high expression levels of receptor transcripts in the protoscolex stage is coincident with the strong immunofluorescent signal detected in the same stage. The restricted expression in the larval protoscolex stage is interesting because this receptor could be used in the future as a reliable marker of cyst fertility. In a previous study [56], different transcriptional levels of the serotonergic GPCR Emu-5-HT$_{7a}$ (hypothetical ortologue of Eca-5-HT$_{7a}$ [10] and described as the gene model EmuJ_001171200 in the publication) was reported between different stages in *E. multilocularis* suggesting a role of this type of receptors in parasite development. In a recent report [57], it was found that serotonin has profound effects in *E. multilocularis* development. It could be interesting to hypothesize if some or all of the developmental effects observed on this last report could be mediated by the serotonergic activation of a putative Emu-5-HT$_{1a}$ (the prescence of an hypothetical *E. multilocularis* version of the receptor deorphanized here) and/ or the activation of Emu-5-HT$_{7a}$ and/or Emu-5-HT$_{7b}$ (the presence of hypothetical *E. multilocularis* versions of the cestode receptors already described in our previous report [10]).

The putative 5-HT1 cestode receptors are probably evolutionary distant orthologues of the human Hsa-5-HT$_{1b}$ receptor. In *H. sapiens*, this latter receptor couples to G protein alpha subunits G$\alpha_i$ or G$\alpha_o$ and is widely expressed in the brain and the cardiovascular system. In the CNS, the Hsa-5-HT$_{1b}$ receptor functions as an inhibitory presynaptic receptor to modulate the release of 5-HT and many other neurotransmitters [58, 59]. The Hsa-5-HT$_{1b}$ receptor is a primary molecular target for the antimigraine drugs ergotamine and dihydroergotamine, which

are efficacious Hsa-5-HT$_{1b}$ receptor agonists. The possibility of repurposing any of the approved drugs for human treatment of parasitic diseases should be evaluated.

In summary, the results of this work suggest that the newly discovered cestode receptor belongs to the 5-HT1 type, is functional responding specifically to serotonin but not to other ligands and could have a major role in the nervous system of these parasites. We believe that this novel type of receptors could be an amenable target for a new generation of cestocidal drugs.

## Supporting information

**S1 Fig. Antigen analysis by western blot and antibody titer evaluation by ELISA.** A) Lane 1, GenScript protein marker (Cat. No. M00521); lane 2, 5 μg of Eca-5-HT$_{1a}$ICL3. The antigen was run in polyacrylamide gel electrophoresis and transferred to a nitrocellulose membrane. The membrane was probed with antibody against poly-histidine tag. The arrow shows the position of the band detected. B) The recombinant antigen produced was administered in six mice and then anti-Eca-5-HT$_{1a}$ICL3 antibody titers were evaluated by ELISA technique. End-point titer was calculated as the reciprocal of the dilution with an Absorbance 450 nm = 0.5. The antibody titers are indicated in the table from the right. "Pre" represents the specific antibody titer of pre-immunized mice.
(TIF)

**S2 Fig. Transcriptional expression levels of serotonergic GPCRs genes in *Echinococcus granulosus*.** Egr-5-HT$_{1a}$, Egr-5-HT$_{1b}$, Egr-5-HT$_{7a}$ and Egr-5-HT$_{7b}$ GPCRs transcriptional expression levels are shown as RPKM (Reads Per Kilobase Million). Comparison of gene expression levels, determined by RNAseq, in several developmental stages of *Echinococcus granulosus sensu stricto* (G1): Oncosphere, Cyst, Protoscolex and Adult [26]. The closest ortho-logue of the receptor studied here, Egr-5-HT$_{1a}$, was marked in bold and the level of transcript expression of this receptor in the protoscolex stage was marked with an arrow. With exception of the protoscolex larval stage, no transcript expression was observed for this receptor in other stages of the parasite.
(TIF)

**S3 Fig. Bioinformatic analyses of the hypothetical Eca-5-HT$_{1b}$ receptor.** A. The bidimensional structure representation and prediction of residues of potential N-glycosylation of Eca-5-HT$_{1b}$ were obtained with the Protter program (http://www.enzim.hu/hmmtop/index.php). The intracellular and extracellular loops are indicated as ICL and ECL respectively. The third intracellular loop used for antibody generation is shaded in grey. Residues potentially involved in N-linked glycosylation were marked in green. B. The amino acid sequences of predicted serotonin receptors ortologues with best scores in blast searches with the cestode Eca-5-HT$_{1b}$ were aligned using the ClustalW method. The new 5-HT1 type cloned receptors' names are marked in bold. The transmembrane (TM), intracellular (ICL) and extracellular (ECL) domains are indicated above each alignment. For the sake of simplicity, the amino terminal end, the transmembrane domain 1, the intracellular loop three and the carboxy terminal end were trimmed partially or completely. The position of residues involved in G protein coupling are indicated with asterisks below each alignment. Residues present in the new predicted receptors that were not seen in other GPCRs are underlined. Critical residues involved in ligand binding and receptor function were indicated in bold below each alignment. Cysteine residues potentially involved in disulphide bond formation are marked as S-S between cysteines. The receptor names, identification numbers and the corresponding species from which the receptors were obtained are enlisted in Table C from S1 Text.
(TIF)

**S4 Fig. Structural comparative analysis between serotonergic type 1 and type 7 receptors.** Comparative analysis of the general structures of the homology models to 5-HT1- vs 5-HT7-type serotonergic receptors from *Echinococcus canadensis* G7 (A) Eca-5-HT$_{1a}$ and (B) Eca-5-HT$_{7a}$; and *Homo sapiens* (C) Hsa-5-HT$_{1a}$ and (D) Hsa-5-HT$_{7a}$. In all the representations, the transmembrane domains I to VII (7-TM) were represented as cartoon while the intracellular loop 3 (ICL3) and C-terminus regions (C-term) were represented as surface. Note the smaller ICL3 and longer carboxy terminal end in 5-HT7-type receptors with respect to 5-HT1-type receptors, in which a bigger ICL3 and smaller C-term (marked as a short cartoon) can be seen.
(TIF)

**S1 Table. G-protein coupling selectivity score for experimentally validated invertebrate 5-HT GPCRs.**
(DOCX)

**S1 Data. Excel spreadsheet containing, in separate sheets, the underlying numerical data for Fig 3A–3D, Fig 4A and 4B, and S1 Fig.**
(XLSX)

**S1 Raw image. Original image from the western blot provided by Genscript in the certificate of analysis of recombinant Eca-5-HT$_{1a}$ICL3.**
(TIF)

**S1 Text. Nucleotide and translated amino acid sequences of the third loop of Eca-5-HT$_{1a}$.**
(DOCX)

**S2 Text. Parameters and Ramachandran plots for the structure models generated in this work.**
(DOCX)

# Acknowledgments

We would like to thanks to Eduardo Gimenez for his technical assistance in the lab. We especially thanks to Doctors Andrés Rossi from the Leloir Institute and Pablo Pomata from the IBYME institute for their special technical skills with the confocal microscope.

# Author Contributions

**Conceptualization:** Federico Camicia, Sang-Kyu Park, Jonathan S. Marchant, Mara C. Rosenzvit.

**Data curation:** Federico Camicia, Hugo R. Vaca, Sang-Kyu Park, Uriel Koziol, Ana M. Celentano, Jonathan S. Marchant, Mara C. Rosenzvit.

**Formal analysis:** Federico Camicia, Hugo R. Vaca, Sang-Kyu Park, Ana M. Celentano, Jonathan S. Marchant, Mara C. Rosenzvit.

**Funding acquisition:** Jonathan S. Marchant, Mara C. Rosenzvit.

**Investigation:** Federico Camicia, Hugo R. Vaca, Sang-Kyu Park, Augusto E. Bivona, Ariel Naidich, Matias Preza, Uriel Koziol, Ana M. Celentano, Jonathan S. Marchant.

**Methodology:** Federico Camicia, Hugo R. Vaca, Sang-Kyu Park, Jonathan S. Marchant, Mara C. Rosenzvit.

**Project administration:** Federico Camicia, Mara C. Rosenzvit.

**Resources:** Jonathan S. Marchant, Mara C. Rosenzvit.

**Supervision:** Federico Camicia, Uriel Koziol, Jonathan S. Marchant, Mara C. Rosenzvit.

**Validation:** Federico Camicia, Hugo R. Vaca, Sang-Kyu Park, Uriel Koziol, Ana M. Celentano, Jonathan S. Marchant, Mara C. Rosenzvit.

**Visualization:** Federico Camicia, Hugo R. Vaca, Sang-Kyu Park, Ana M. Celentano, Jonathan S. Marchant, Mara C. Rosenzvit.

**Writing – original draft:** Federico Camicia.

**Writing – review & editing:** Federico Camicia, Hugo R. Vaca, Sang-Kyu Park, Augusto E. Bivona, Ariel Naidich, Uriel Koziol, Ana M. Celentano, Jonathan S. Marchant, Mara C. Rosenzvit.

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
