## [Decision Letter · Decision Letter 0]

8 May 2021

PONE-D-21-12202

Characterization of a new type of neuronal 5HT G protein coupled receptor in the cestode nervous system

PLOS ONE

Dear Dr. Camicia,

Thank you for submitting your manuscript to PLOS ONE. After careful consideration, we feel that it has merit but does not fully meet PLOS ONE’s publication criteria as it currently stands. Therefore, we invite you to submit a revised version of the manuscript that addresses the points raised during the review process

After reading the manuscript and the Reviewer's comments for PLOS Neglected Tropical Diseases, I decided to have the manuscript reviewed by an expert in the field of GPCRs of invertebrates (Reviewer #1). The three Reviewers in the first round were all parasitology experts. Since I myself have some experience with characterizing serotonin receptors, I absolutely agree with the criticisms made by Reviewer #1. In order to demonstrate an inhibitory effect on adenylate cyclase, it is essential to first stimulate the enzyme with forskolin or NKH447. A number of potential 5HT receptor antagonists should be tested, not just those that are highly specific for the 5-HT1A subtype in vertebrates. For the selection of the ligands, I strongly recommend a look at the insect literature (Drosophila melanogaster, Apis mellifera, Periplaneta amercicana, Tribolium castaneum ...). For more details, please see the report of Reviewer #1. For your information: Reviewer #2 corresponds to Reviewer #3 of the 1st round).

We look forward to receiving your revised manuscript.

Kind regards,

Wolfgang Blenau

Academic Editor

PLOS ONE

Journal Requirements:

2. In your Methods section, please provide the name of the slaughterhouse where the animals were sacrificed.

Reviewers' comments:

Reviewer's Responses to Questions

**Comments to the Author**

1. Is the manuscript technically sound, and do the data support the conclusions?

Reviewer #1: Partly

Reviewer #2: Yes

2. Has the statistical analysis been performed appropriately and rigorously? 

Reviewer #1: Yes

Reviewer #2: Yes

3. Have the authors made all data underlying the findings in their manuscript fully available?

Reviewer #1: Yes

Reviewer #2: Yes

4. Is the manuscript presented in an intelligible fashion and written in standard English?

Reviewer #1: Yes

Reviewer #2: Yes

5. Review Comments to the Author

Reviewer #1: Camicia et al. provide a revised version of their manuscript that I had not evaluated in the first round of the reviewing process. The authors have conducted a molecular and bioinformatics approach and identified potential members of the serotonin (5-HT) 1 clade of G-protein coupled receptors in E. canadensis, M. vogae, and H. microstoma. Classifiction of the receptor proteins is mainly based on bioinformatic evidence. The manuscript in general is valuable and extends current knowledge on the repertoire of biogenic amine receptors expressed in platyhelminthes. The bioinformatics part of study has been conducted with high precision and depth. However, certain aspects especially the pharmacological characterization of the receptors require further experimental investigation to unequivocally support classifying the receptors as 5-HT1 clade members. My general and specifc comments are outlined in the following paragraphs.

General comments

Throughout the manuscript the authors should strictly use the conventional nomenclature used by the ‘serotonin receptor community’: serotonin (= 5-hydroxy tryptamine, 5-HT) and subscript for receptor sub-families: 5-HT1.

Another fail in spelling should be corrected: use G-protein coupled receptors (GPCRs) as well as G-proteins throughout the manuscript.

Substitute ‘dose(s)’ for concentration(s); and use the term ‘concentration-response curve’.

A main issue of concern, however, goes to the pharmacological data provided in the manuscript. It is a common and well-taken approach to study, e.g., a newly cloned receptor gene in a ‘reporter’ cell line. For this purpose the authors have chosen cells that constitutively express a promiscuous G-protein (G15). This is a smart experimental design to gain first insight into a receptor’s functionality. However, the data shown in Fig. 3A and 3B raise certain doubts and are (in part) incorrect:

a) in Fig. 3A the concentration is missing. I guess it is µM(?).

b) scaling of the ordinate is inappropriate (omit negative values; in 3B, too) and has to be changed.

c) exposure/measuring time for cell-based Ca++ fluorescent assays would be extremely long according to the values given on the abscissa (approx. 80 min total measuring time). In the Mat. and Meth. section, the authors state that measurements were performed in scales of seconds. What is correct? I assume, as typical for Ca++ imaging experiments, the latter and, thus, the graph requires editing.

d) The authors state that Fig.3A contains raw data. This does not fit to Mat. and Meth. (l.337ff) and it would be better to display the graphs either as �RFU vs time or as �RFU/R0 (= mean basal fluorescence) vs time.

e) How do the authors explain that fluorescent signals decrease approx. 40 min (sec?) after 5-HT application for almost all ligand concentrations, except for 50µM?

f) Which concentrations (5-HT) were used to calculate the concentration-response curve displayed in Fig.3B? Labeling of the ordinate seems to be wrong (5-HT µM). I guess it is (M), otherwise the deduced EC50 would be six orders of magnitude smaller than given in the manuscript (~7.7 nM). Data points for (0µM) 5-HT are missing! Have they been assigned to 10-11µM 5-HT? This should be changed and the graph adapted accordingly. Furthermore, instead of 100µM (=10-4M) 5-HT, the highest concentration of 5-HT shown in Fig.3A and also stated in Mat. and Meth. was 50µM (=5x10-4M). Therefore, the graph has to be corrected also in this respect.

g) Fig.3C should be exchanged for a bar graph in which the normalized peak values of fluorescence are displayed and, eventually, negative values might be defined as 0. It would have been also more appropriate to use identical ligand concentrations (here 10 µM) in all measurements and the authors should provide these data. Can the authors explain why histamine application causes a fluorescent signal?

h) The experiments performed to examine the potential coupling of the receptor to cAMP signalling suffer from faulty experimental design. Given that the receptor belongs to the 5-HT1 subfamily, the most obvious assumption were, that receptor activation leads to a reduction of the intracellular cAMP concentration. To assess this effect experimentally, cells have to be incubated with (1) IBMX to block PDE activities and (b) with forskoin or its derivative (NKH477) to stimulate endogenous membrane-bound adenylyl cyclases. In this way, one would gain an increase in cAMP. Upon adding a concentration series of 5-HT, the authors should be able to register a reduction of the cAMP concentration – leading to a nice and convincing demonstration of the correct assignment of the receptor. The authors may employ their (cAMP)-reporter cell assay referred to in the manuscript for this purpose or use a commercial kit to determine cAMP concentrations. Without such data, the value of the manuscript is considerably reduced.

i) A final concern addresses the pharmacological profile of the receptor. The authors state that potential agonists did not or only scarcely induce receptor activity. However, the authors should examine and provide data on the effects of antagonists, too. They may select characteristic ones from the rich source of compounds that are known to interact with 5-HT1 receptors in a variety of other species, including insects. It should be rather straightforward to use the Ca++ imaging set-up to perform such experiments.

Specific comments

l.38: ‘…GPCR motifs…’

l.40: as stated above change nomenclature to Eca-5-HT1a

l.88: change to: ‘…5-HT receptors are known but…’

l.90f: skip ‘…determined by similarities in sequence and transduction mechanisms [ ]’

l.92: ‘…system exists based…’

l.92-94: referring to C.elegans assignment of 5-HT clades is more confusing than helpul. Skip sentence.

l.95ff: re-order sentences; starting with 5-HT1 receptors and ending with 5-HT7-like receptors.

l.104f: redundant information; skip ‘…[17, 18] but no GPCRs…..In nematodes…’

l.109: introduce ‘…human (homo sapiens) Hsa5-HT1b …’

l.115-117: redundant to information given in lines 110-114; skip

l.152f: the authors state that they prepared working dilutions of 5-HT in RPMI medium. Later they state that ligands were applied in assay buffer. Check and correct accordingly.

l.274: ‘…cloning…’ has been achieved after PCR amplification of cDNA fragments but not sequencing!

l.295: ‘…BLAST searches…’

l.299: ‘…denaturation…’

l.302: final extension for 2 mins?

l.311ff: shorten and rephrase para: e.g. ‘The cloned cDNA from E. canadensis was named Eca-5-HT1a….’

l.351ff: explain in more detail the assay conditions. What is a ‘G�s mutant’? For what purpose is it applied? How does it enter cells? Was IBMX used? Was Forskolin applied? Did the authors really add 500mM Na butyrate?

l.457: skip ‘…the mRNA or transcript…’

l.458: change to ‘…encoding the E. granulosus orthologue of this receptor…’

l.467: change to ‘…within the 5-HT1 clade (Fig. 1A) were named Eca-5-HT1a,…’

l.471: change to ‘…receptors identified in this work’

Figure1 and Figure legend:

There should be no N-glycosylation sites labelled in ICL3!

Why was TM1 excluded from the alignment?

The authors should give names and accession numbers of all receptor sequences in the Fig. legend.

l.473: change to ‘The structural representation…’

l.496ff: skip (clades 4, 7, 1) from legend; misleading because the authors only show three branches. Did the authors use an outgroup for calculating the phylogenetic tree?

l.509: skip ‘Overall…’

l.511: skip ‘…(like 5-HT or other ligands),…’

l.512: change to ‘The new…grouped within the 5-HT1 clade of serotonergic receptors in the phylogenetic tree (Fig. 1C).’

l.520: change to ‘Similar to Eca-5-HT1a, multiple sequence alignments…’

l.522: change to 61.2 kDa

l.526: change to ‘…within the 5-HT1 clade…’

l.530ff: change to: ‘However, this annotated receptor was not examined experimentally.’

l.533: change to ‘For labelling important aa residues, the Ballesteros…’

l.547: change to ‘Table 1 summarizes…’

l.549: change to ‘Corresponding residues in invertebrate…’

l.555f: skip ‘and classification’

l.570: change to ‘…5-HT2 type receptors.’

l.575: change to ‘ the Eca-5-HT1b receptor harbors a serine…’

l.580f: change to ‘The presence of threonine at position 5.39, a residue…’

l.590ff: change to ‘In the PIF motif…., the isoleucine (..) is replaced by a threonine in all three cloned…’

l.601: change to ‘Residues potentially involved in receptor and G-protein coupling’

l.613: change to ‘…GPCR groups…’

l.620ff: Table 3 is non-informative at all and should skipped. Spacing between residues located in different TMs is strongly influenced by the lengths of intervening loops. The only interesting aspect one might gain, is that many Gi-coupled receptors possess short C-termini, a feature also seen for the cestode receptors.

l.637f: phrase more patiently, a bioinformatics analysis of potential interacting G-protein subtypes should not be over-emphasized. For all receptors decent interaction with Gi-proteins is evident.

l.655: change to ‘…have a similar structure as Hsa-5-HT1b…’

l.701: give aa residues found at indicated positions

l.703: change to ‘The isoleucine at position…’

l.710ff: substitute’big’ for long

l.715ff: skip whole para; Eca-5-HT1b has not been examined experimentally in the contribution and the para duplicates results already summarized for the other receptors.

l.732: change to: ‘Addition of serotonin to cells transfected with the Eca-5-HT1a construct induced calcium release from intracellular stores in a concentration dependent manner (Fig…)’

l.737: change to ‘Heterologous expression of Eca-5-HT1a receptors’

l.738ff: Figure legend must be rephrased once corrections and additional experiments were included in Fig.3

l.756ff: statement is wrong and misleading! See my general comments. The authors must provide new data on the receptor : G-protein interaction. G15 is not a proper control!

l.760: skip ‘…(neurotransmitters…others)…’

l.761: Histamine also leads to an increase of the Ca++ dependent signal. Please explain!

l.762ff: whole para is not convincing as outlined in my general comments. The authors have to address the ability of Eca-5-HT1a to inhibit cAMP production by a series of novel experiments.

L768f: Misleading statement! So far, the authors have only in-appropriately examined coupling of Eca-5-HT1a to other G-proteins. As it reads, even in E. canadensis, the receptor would require G15 as an interacting partner to become functional.

l.775-781: rephrase and skip all informations already given in Mat. and Meth.

l.800: ‘…shown in F…’

l.807: ‘…shown in H…’

l.836f: change to ‘…receptor-containing fibers could receive serotonergic…’

l.847: change to: ‘…assignment the proteins as a 5-HT1 type receptors…’

l.851: skip info about spacing between TMs

l.858: change to ‘…is expected to be coupled…’

l.862ff: statement concerning coupling to Gi-proteins not sufficiently examined (s.a.); skip

l.887-910: whole para suffers from the in-appropriately executed experiments! Conclusions are not supported by current data.

l.916: change to: ‘…images shown…to release serotonin that could be…’

l.923: change to: ‘…low titers and high concentrations of the serotonin…’

l.936: change to: ‘…Sm-5-HTR was expressed…’

l.946f: skip ‘…, a serotonergic receptor…[57].’

l.951-956: shorten para because until now UCM2550 has not been proven to be active at Eca-5-HT1a.

l.961: change to ‘…receptor transcripts…’

l.963: change to: ‘…expression in the…stage is interesting…’

l.970ff: rephrase last sentence – strengthen wording

l.980: change to ‘…repurposing any of the approved drugs…’

l.1344: change to ‘…GPCR genes…’; most likely heading will be no more required when moving receptor names and accession numbers to legend of Fig.1.

l.1266: change to ‘Comparison of gene expression levels, determined…’

Figure S2: skip labelling of N-glycosylation sites in ICL3

Figure S4: skip completely

Figure S5: skip completely; seriously, part D does not fit to the data shown in the main text!

Figure S6: skip completely; as mentioned above, the data were obtained under in-appropriate conditions

Reviewer #2: This manuscript describes the detailed structural analysis and functional characterisation of a new type of cestode 5HT receptor. Multiple different methods were used to characterise the receptor including in silico analysis, expression in HEK cells and immunolocalization. Authors concluded that the 5HT receptor is likely be classified as type 1.

The real strengths of this manuscript lie in the comprehensive structural analysis of the receptors identified with very strong data indicating type 1 characterisation. The authors have appropriately addressed technical difficulties common with functional expression of receptors in HEK cells and provided more than sufficient data to prove that the receptor functions as a serotonin receptor. Although authors could not definitively functionally characterise this 5HT receptor as type 1, this is reflected appropriately in the final conclusion of the study and the language used when describing the receptor. The results have been condensed and presented in a very clear format highlighting the most important data within the study. Immunolocalization images are easy to interpret showing appropriate expression patterns for the conclusions which have been drawn from the study. Although background staining using the serotonin antibody is obvious, the authors have addressed this issue within the manuscript/review process and the staining presented for the 5HT receptor antibody is very clear and much more defined. Combined serotonin and 5HT receptor staining is acceptable for the conclusions drawn using this data.

Overall I believe this manuscript should be accepted for publication as the authors have made an impressive effort to address all issues/concerns from reviewers and really focussed the details of the study. The data presented are accurate supporting the discussions/conclusions and the manuscript has been edited to a very high standard.

6. PLOS authors have the option to publish the peer review history of their article (what does this mean?). If published, this will include your full peer review and any attached files.

Reviewer #1: No

Reviewer #2: No

---

## [Author Response · Author response to Decision Letter 0]

16 Sep 2021

September 16, 2021

Dear Editor and Reviewers, 

We are grateful for all your constructive input into our manuscript and will, in the following pages, provide answers to your questions and concerns. A revised manuscript, including tracked changes, as well as a final version, are attached to the submission. 

Sincerely yours, 

Dr Mara C. Rosenzvit 

IMPAM-UBA-CONICET, Facultad de Medicina - Universidad de Buenos Aires Paraguay 2155, piso 13 (1121) Buenos Aires – ARGENTINA

Tel: (5411) 5950-9500 ext. 2192 Fax: (5411) 5950-9577

E-mail: mrosenzvit@fmed.uba.ar; mararosenzvit@gmail.com

Professor Dr. Jonathan S. Marchant 

Marcus Professor & Chair

Department of Cell Biology, Neurobiology & Anatomy 8701 Watertown Plank Road

Post Office Box 26509

Milwaukee, Wisconsin 53226-0509

Tel: (414) 955-8261

Fax: (414) 955-6517

jmarchant@mcw.edu

¬¬¬¬¬¬¬¬¬¬¬¬¬¬¬¬¬¬¬

Reviewers' comments:

Reviewer's Responses to Questions

Comments to the Author

1. Is the manuscript technically sound, and do the data support the conclusions?

Reviewer #1: Partly

Reviewer #2: Yes

2. Has the statistical analysis been performed appropriately and rigorously?

Reviewer #1: Yes

Reviewer #2: Yes

3. Have the authors made all data underlying the findings in their manuscript fully available?

Reviewer #1: Yes

Reviewer #2: Yes

4. Is the manuscript presented in an intelligible fashion and written in standard English?

Reviewer #1: Yes

Reviewer #2: Yes

5. Review Comments to the Author

Reviewer #1: Camicia et al. provide a revised version of their manuscript that I had not evaluated in the first round of the reviewing process. The authors have conducted a molecular and bioinformatics approach and identified potential members of the serotonin (5-HT) 1 clade of G-protein coupled receptors in E. canadensis, M. vogae, and H. microstoma. Classifiction of the receptor proteins is mainly based on bioinformatic evidence. The manuscript in general is valuable and extends current knowledge on the repertoire of biogenic amine receptors expressed in platyhelminthes. The bioinformatics part of study has been conducted with high precision and depth. However, certain aspects especially the pharmacological characterization of the receptors require further experimental investigation to unequivocally support classifying the receptors as 5-HT1 clade members. My general and specifc comments are outlined in the following paragraphs.

General comments

Throughout the manuscript the authors should strictly use the conventional nomenclature used by the ‘serotonin receptor community’: serotonin (= 5-hydroxy tryptamine, 5-HT) and subscript for receptor sub-families: 5-HT1.

Another fail in spelling should be corrected: use G-protein coupled receptors (GPCRs) as well as G-proteins throughout the manuscript.

Substitute ‘dose(s)’ for concentration(s); and use the term ‘concentration-response curve’.

Thank you very much for this feedback. We have included the conventional nomenclature to refer to serotonin all along the manuscript. We also included the subscript for each receptor sub-family. We have also included the terms “G-protein coupled receptor” as well as “G-proteins” in the manuscript. The term “dose(s)” was replaced by “concentration(s)”. Finally, we included the term “concentration-response curve”. 

A main issue of concern, however, goes to the pharmacological data provided in the manuscript. It is a common and well-taken approach to study, e.g., a newly cloned receptor gene in a ‘reporter’ cell line. For this purpose the authors have chosen cells that constitutively express a promiscuous G-protein (G15). This is a smart experimental design to gain first insight into a receptor’s functionality. However, the data shown in Fig. 3A and 3B raise certain doubts and are (in part) incorrect:

a) in Fig. 3A the concentration is missing. I guess it is µM(?).

Thank you for spotting this error. Yes, the concentration is in µM and this was changed in the new revised graph.

b) scaling of the ordinate is inappropriate (omit negative values; in 3B, too) and has to be changed.

The ordinates from figures 3A and 3B have been changed as requested. 

c) exposure/measuring time for cell-based Ca++ fluorescent assays would be extremely long according to the values given on the abscissa (approx. 80 min total measuring time). In the Mat. and Meth. section, the authors state that measurements were performed in scales of seconds. What is correct? I assume, as typical for Ca++ imaging experiments, the latter and, thus, the graph requires editing.

We didn´t realize about this mistake in our previous submission. The measurements were performed in scales of seconds and the graphs have been edited accordingly.

d) The authors state that Fig.3A contains raw data. This does not fit to Mat. and Meth. (l.337ff) and it would be better to display the graphs either as �RFU vs time or as �RFU/R0 (= mean basal fluorescence) vs time.

The data represent raw fluorescence units, and we feel there is merit in presenting raw data. ΔF/Fo is approximately 12 for maximal responses (to 5-HT) under our assays conditions with fluo-4 as the reporter. 

e) How do the authors explain that fluorescent signals decrease approx. 40 min (sec?) after 5-HT application for almost all ligand concentrations, except for 50µM?

Thank you for this observation. As this reviewer states, the fluorescent signal of almost all ligand concentrations decrease after 40 seconds, even for peak 5-HT concentrations (50µM). For example, see the following graph: at longer times, it can be seen that also, at 50 µM of 5-HT the fluorescent signal decreases too. This is because in intact cells, elevated cytoplasmic Ca2+ is removed from the cell by extrusion across the cell surface/endoplasmic reticulum membranes away from the cytosolic reporter. As the reviewer may be implying this could well imply a desensitization in the cestode GPCR responsiveness to maximal 5-HT, as observed in many GPCR systems. Further studies would be needed to investigate the molecular basis underpinning this effect. 

Figure 1: Time resolved measurements of Ca2+ accumulation (measured as change in relative fluorescence units, ΔRFU) in cells expressing the cestode Eca-5-HT1a receptor before and after addition of different concentrations of 5-HT (arrow, concentrations indicated in legend).

f) Which concentrations (5-HT) were used to calculate the concentration-response curve displayed in Fig.3B? Labeling of the ordinate seems to be wrong (5-HT µM). I guess it is (M), otherwise the deduced EC50 would be six orders of magnitude smaller than given in the manuscript (~7.7 nM). Data points for (0µM) 5-HT are missing! Have they been assigned to 10-11µM 5-HT? This should be changed and the graph adapted accordingly. Furthermore, instead of 100µM (=10-4M) 5-HT, the highest concentration of 5-HT shown in Fig.3A and also stated in Mat. and Meth. was 50µM (=5x10-4M). Therefore, the graph has to be corrected also in this respect.

Thank you very much for this observation. The labeling of the ordinate was wrong and changed to Molar (M). Data points for zero concentrations were now included in the graph. The highest concentration of 5-HT for the new revised graph was 100 µM (= 1 x 10-4 M) as mentioned. The graph was changed following the suggestions made by the reviewer.

g) Fig.3C should be exchanged for a bar graph in which the normalized peak values of fluorescence are displayed and, eventually, negative values might be defined as 0. It would have been also more appropriate to use identical ligand concentrations (here 10 µM) in all measurements and the authors should provide these data. Can the authors explain why histamine application causes a fluorescent signal?

We have changed Figure 3C as suggested and have also incorporated parallel measurements in control cells (not expressing the GPCR). For symmetry we have now included identical concentrations of all ligands (10uM). 

As can been seen in the new data, the tiny responsiveness to histamine is also seen in control HEK293-GNA15 cells, suggesting the expression of endogenous histamine receptors in the HEK293-GNA15 cell line. For this reason, the fluorescent signal observed initially with histamine not seems to be relevant in the study of the Eca-5-HT1a receptor specificity.

h) The experiments performed to examine the potential coupling of the receptor to cAMP signalling suffer from faulty experimental design. Given that the receptor belongs to the 5-HT1 subfamily, the most obvious assumption were, that receptor activation leads to a reduction of the intracellular cAMP concentration. To assess this effect experimentally, cells have to be incubated with (1) IBMX to block PDE activities and (b) with forskoin or its derivative (NKH477) to stimulate endogenous membrane-bound adenylyl cyclases. In this way, one would gain an increase in cAMP. Upon adding a concentration series of 5-HT, the authors should be able to register a reduction of the cAMP concentration – leading to a nice and convincing demonstration of the correct assignment of the receptor. The authors may employ their (cAMP)-reporter cell assay referred to in the manuscript for this purpose or use a commercial kit to determine cAMP concentrations. Without such data, the value of the manuscript is considerably reduced.

Based on the reviewer’s feedback, we have removed these experiments from the manuscript as these negative data do not add to the narrative. We tried to demonstrate changes in cAMP signaling using the cADDis system, but were unsuccessful. The methodology however was appropriate to elevate cAMP, as a constitutively active Gαs was introduced into cells (as per manufacturer’s optimization) (1, 2). 

Instead, we now provide new data with a suite of point mutations around the 5-HT binding pocket that provide further evidence that mutation of residues known to coordinate 5-HT ablate the function of the cestode GPCR (new Figure 3D, lines 1174-1205 of the new revised version with marked changes). Further research into the mechanisms of parasite GPCR/mammalian G proteins are needed to solve the problem of coupling. 

Given that we were unable to experimentally confirm if Eca-5-HT1a is indeed 5-HT1 type of receptors, we moderated the strength of the affirmations/conclusions related to receptor classification in the abstract, results and discussion sections. We employed the terms “likely”, “presumptive” and/or “putative” all along the article when we talk about receptor assignment to a specific clade. Coupling specificity is only suggested here based on receptor´s phylogenetic and structural data. 

However, based in phylogenetics, alignments and modeling studies we hypothesize here the existence of receptors belonging to another type to those already described in our previous publication (3). We believe that the findings reported here (in case they were disseminated) will open the door and will encourage to other researchers to search new receptors in this phylum. The lack of coupling observed here (between the cestode receptor and the mammalian Galphai/o protein subunit), far from being uninteresting could instead mean a different (and very interesting) mechanism of interaction between the cestode receptor and their native G alpha i/o coupling partner. 

In the context of the experimental pharmacology in parasites, in which novel targets and drugs are urgently needed to treat zoonotic diseases (like cystic echinococcosis for example) and the absolute lack of information about this type of receptors in this phylum (where only 5-HT7 receptors were described so far), the structural peculiarities of the newly discovered receptor, and more interesting, the localization of this receptor in the nervous system of the parasite (perhaps well known in other invertebrate species like insects but not in Platyhelminths) are key aspects that should be further considered for publication. The situation is very different for example in the context of the insect receptor biology in which many receptors of the 5-HT1 clade and other major families of serotonin receptors where already described. 

References:

1-Manual, catalog number #X0200G from Montana Molecular. https://montanamolecular.com/product/x0200g-green-gi-caddis-camp-assay-kit/

2-Tewson P, Martinka S, Shaner N, Berlot C, Quinn AM, Hughes T. Assay for Detecting Gαi-Mediated Decreases in cAMP in Living Cells. SLAS Discov. 2018 Oct;23(9):898-906. 

3- Camicia F, Celentano AM, Johns ME, Chan JD, Maldonado L, Vaca H, Di Siervi N, Kamentezky L, Gamo AM, Ortega-Gutierrez S, Martin-Fontecha M, Davio C, Marchant JS, Rosenzvit MC. Unique pharmacological properties of serotoninergic G-protein coupled receptors from cestodes. PLoS Negl Trop Dis. 2018 Feb 9;12(2):e0006267. 

i) A final concern addresses the pharmacological profile of the receptor. The authors state that potential agonists did not or only scarcely induce receptor activity. However, the authors should examine and provide data on the effects of antagonists, too. They may select characteristic ones from the rich source of compounds that are known to interact with 5-HT1 receptors in a variety of other species, including insects. It should be rather straightforward to use the Ca++ imaging set-up to perform such experiments.

In the new revised version of the manuscript, we have included an entire new section devoted to Eca-5-HT1a pharmacology (please see lines 1207-1231 of the new revised version with marked changes). During the course of the new experiments performed, we have found by chance a new 5-HT GPCR agonist called tryptamine. The response to tryptamine seems to be strong but lower than to 5-HT (Fig 3 C). The agonist activity observed in Eca-5-HT1a resulted interesting since tryptamine was shown inactive in the insect Trica5-HT receptor (an invertebrate 5-HT1 GPCR described in the red flour beetle (Tribolium castaneum) suggesting interesting pharmacological differences between invertebrate receptors (1). 

The 5-HT GPCRs agonists 5-methoxytryptamine and α-methylserotonin displayed concentration-response activity in a stable cell line expressing the Eca-5-HT1a receptor (Fig 4A). 

Moreover, we have found that methysergide display strong inhibitory activity using the same stable cell line (Fig 4B). This compound has shown inhibitory activity in other invertebrate 5-HT1 receptors, e.g. Trica5-HT suggesting that Eca-5-HT1a could be a 5-HT1 receptor type (1).

1-Vleugels R, Lenaerts C, Baumann A, Vanden Broeck J, Verlinden H (2013) Pharmacological Characterization of a 5-HT1-Type Serotonin Receptor in the Red Flour Beetle, Tribolium castaneum. PLoS ONE 8(5): e65052. https://doi.org/10.1371/journal.pone.0065052

Specific comments

l.38: ‘…GPCR motifs…’

Suggestion included in the text, line 57.

l.40: as stated above change nomenclature to Eca-5-HT1a

The original name of the receptor (Eca-5HT1a) was changed for (Eca-5-HT1a) in all the manuscript, figures and supplementary material as suggested. 

l.88: change to: ‘…5-HT receptors are known but…’

Suggestion included in the text, line 117.

l.90f: skip ‘…determined by similarities in sequence and transduction mechanisms [ ]’

Suggestion included in the text, line 119.

l.92: ‘…system exists based…’

Removed with the sentence, line 120.

l.92-94: referring to C.elegans assignment of 5-HT clades is more confusing than helpul. Skip sentence.

Removed as suggested, line 120.

l.95ff: re-order sentences; starting with 5-HT1 receptors and ending with 5-HT7-like receptors.

The sentences were re-ordered as suggested, see lines 120 to 140.

l.104f: redundant information; skip ‘…[17, 18] but no GPCRs…..In nematodes…’

Removed as suggested, line 144.

l.109: introduce ‘…human (homo sapiens) Hsa5-HT1b …’

Introduced as suggested, line 148.

l.115-117: redundant to information given in lines 110-114; skip

Removed from the text as suggested, line 154.

l.152f: the authors state that they prepared working dilutions of 5-HT in RPMI medium. Later they state that ligands were applied in assay buffer. Check and correct accordingly.

Working dilutions were prepared in RPMI medium. The word “assay buffer” was changed by “RPMI” in line 449.

l.274: ‘…cloning…’ has been achieved after PCR amplification of cDNA fragments but not sequencing!

The word sequencing was removed from this sentence in line 373 as suggested.

l.295: ‘…BLAST searches…’

The plural form of the word BLAST was removed as suggested in line 400.

l.299: ‘…denaturation…’

The word was corrected as suggested in line 404.

l.302: final extension for 2 mins?

The right final extension step was of 30 minutes and included in line 407.

l.311ff: shorten and rephrase para: e.g. ‘The cloned cDNA from E. canadensis was named Eca-5-HT1a….’

Shortened and rephrased as suggested in lines 415-416. 

l.351ff: explain in more detail the assay conditions. What is a ‘G�s mutant’? For what purpose is it applied? How does it enter cells? Was IBMX used? Was Forskolin applied? Did the authors really add 500mM Na butyrate?

The assay conditions were the following: cAMP Gi assay was performed in live cells using the green difference detector in situ (cADDis) cAMP sensor (Montana Molecular, Bozeman, MT) (1). HEK‐293 or HEK-293 transiently transfected with Eca-5-HT1a cells were plated on a black‐walled, clear flat bottom 96‐well plates along with 20-25 µl of cADDis sensor, 4 µM Gαs mutant, 500 mM Sodium butyrate in Dulbecco's Modified Eagle Medium with 10% FBS. Alternatively, hD2 receptor was used as a control instead of Eca-5-HT1a. 24 hr after incubation at 5% CO2 and 37°C, Media was aspirated and replaced with the culture media containing 100 µl per well of 1X Dulbecco's Phosphate Buffered Saline Solution with calcium and magnesium. Then, the 96‐well plate was covered with aluminum foil and incubated at room temperature for 30 minutes. Cell fluorescence was read from the plate bottom using 485 nm of excitation and 535 nm of emission wavelengths using a SpectraMax i3X plate reader (Molecular Devices). Basal fluorescence was monitored for 5 minutes, then serotonin was added, and the fluorescence signal (raw fluorescence units) was monitored for an additional 30 minutes with 30 seconds intervals. For quantitative analyses, peak fluorescence was further analyzed in Origin.

The cADDis assay decreases fluorescence intensity when cAMP is increasing in the cell and increases in fluorescence in response to activation of Gαi (1,2). To detect the activation of Gαi, the steady state levels of cAMP should first be increased with the constitutively active Gαs provided in the kit (1,2). The G�s mutant is a constitutively active G alpha protein which increases steady-state levels of cAMP and eliminates the need to add forskolin (3). This gene enters the cell by transduction together the cAMP sensor. The response of the control receptor hD2 observed strongly suggest that the whole experiment was properly performed. 

References:

1-Manual, catalog number #X0200G from Montana Molecular. https://montanamolecular.com/product/x0200g-green-gi-caddis-camp-assay-kit/

2-Tewson P, Martinka S, Shaner N, Berlot C, Quinn AM, Hughes T. Assay for Detecting Gαi-Mediated Decreases in cAMP in Living Cells. SLAS Discov. 2018 Oct;23(9):898-906. 

3-Grishina G, Berlot CH. Identification of common and distinct residues involved in the interaction of alphai2 and alphas with adenylyl cyclase. J Biol Chem. 1997 Aug 15;272(33):20619-26. 

l.457: skip ‘…the mRNA or transcript…’

The indicated words were removed as suggested, line 517.

l.458: change to ‘…encoding the E. granulosus orthologue of this receptor…’

Changed as suggested, line 518.

l.467: change to ‘…within the 5-HT1 clade (Fig. 1A) were named Eca-5-HT1a,…’

Changed as suggested, line 526.

l.471: change to ‘…receptors identified in this work’

Changed as suggested, line 540.

Figure1 and Figure legend:

There should be no N-glycosylation sites labelled in ICL3!

Thank you for this observation. Labels for N-glycosylation sites in ICL3 were removed from the figure.

Why was TM1 excluded from the alignment?

In most of the reviewers about serotonergic GPCRs, the first TM1 is not usually mentioned. In our previous submission, we thought that the inclusion of this domain could disperse or distract the reader about which part of the sequence should first look at. However, this time, we have included this domain in the new version of the figure following reviewer suggestion.

The authors should give names and accession numbers of all receptor sequences in the Fig. legend.

Names and accession numbers of all sequences were included in the new revised version. Lines 541 to 542, 551 to 570, 581 to 591 and 707. 

l.473: change to ‘The structural representation…’

Changed as suggested, line 543.

l.496ff: skip (clades 4, 7, 1) from legend; misleading because the authors only show three branches. Did the authors use an outgroup for calculating the phylogenetic tree?

Clades removed as suggested, line 701-702. We have included an outgroup for calculating the phylogenetic tree. 

l.509: skip ‘Overall…’

Removed as suggested, line 716.

l.511: skip ‘…(like 5-HT or other ligands),…’

Removed as suggested, line 717.

l.512: change to ‘The new…grouped within the 5-HT1 clade of serotonergic receptors in the phylogenetic tree (Fig. 1C).’

Changed as suggested, lines 718-720.

l.520: change to ‘Similar to Eca-5-HT1a, multiple sequence alignments…’

Changed as suggested, lines 725-741.

l.522: change to 61.2 kDa

Changed as suggested: comma replaced by dot, line 742. 

l.526: change to ‘…within the 5-HT1 clade…’

Changed as suggested, lines 745-746.

l.530ff: change to: ‘However, this annotated receptor was not examined experimentally.’

Changed as suggested, line 749.

l.533: change to ‘For labelling important aa residues, the Ballesteros…’

Changed as suggested, line 750.

l.547: change to ‘Table 1 summarizes…’

Changed as suggested, line 782.

l.549: change to ‘Corresponding residues in invertebrate…’

Changed as suggested, line 783-784.

l.555f: skip ‘and classification’

Removed as suggested, line 790.

l.570: change to ‘…5-HT2 type receptors.’

Changed as suggested, line 841.

l.575: change to ‘ the Eca-5-HT1b receptor harbors a serine…’

Changed as suggested, line 846.

l.580f: change to ‘The presence of threonine at position 5.39, a residue…’

Changed as suggested, line 852.

l.590ff: change to ‘In the PIF motif…., the isoleucine (..) is replaced by a threonine in all three cloned…’

Change as suggested, lines 868-870.

l.601: change to ‘Residues potentially involved in receptor and G-protein coupling’

Change as suggested, lines 879.

l.613: change to ‘…GPCR groups…’

Removed with the Table 3.

l.620ff: Table 3 is non-informative at all and should skipped. Spacing between residues located in different TMs is strongly influenced by the lengths of intervening loops. The only interesting aspect one might gain, is that many Gi-coupled receptors possess short C-termini, a feature also seen for the cestode receptors.

Removed as suggested. We think that both: short C-termini and long intracellular loop are major structural aspects characteristic of Gi-coupled receptors. The long intracellular loop is another important structural feature of Gi-coupled receptors that, to the best of our knowledge, should not be underrated.

l.637f: phrase more patiently, a bioinformatics analysis of potential interacting G-protein subtypes should not be over-emphasized. For all receptors decent interaction with Gi-proteins is evident.

We changed and moderated our affirmations, lines 940-941.

l.655: change to ‘…have a similar structure as Hsa-5-HT1b…’

Changed as suggested, line 984.

l.701: give aa residues found at indicated positions

Added as suggested, line 1050.

l.703: change to ‘The isoleucine at position…’

Changed as suggested, line 1052.

l.710ff: substitute’big’ for long

Changed as suggested, line 1058.

l.715ff: skip whole para; Eca-5-HT1b has not been examined experimentally in the contribution and the para duplicates results already summarized for the other receptors.

Removed as suggested.

l.732: change to: ‘Addition of serotonin to cells transfected with the Eca-5-HT1a construct induced calcium release from intracellular stores in a concentration dependent manner (Fig…)’

Changed as suggested, line 1080. 

l.737: change to ‘Heterologous expression of Eca-5-HT1a receptors’

Change as suggested, line 1084.

l.738ff: Figure legend must be rephrased once corrections and additional experiments were included in Fig.3

Change as suggested, line 1084 to 1164. 

l.756ff: statement is wrong and misleading! See my general comments. The authors must provide new data on the receptor : G-protein interaction. G15 is not a proper control!

As we explain before, downward cADDis experiments (permutated probe which responds to cAMP decreases with fluorescence increases) seem to show that the cestode receptor don´t couple with the endogenous mammalian G alpha i/o subunit. We are sure that this could not be attributed to a lack of technical skills since mammalian positive control is responding. 

We already realized that unfortunately, in spite of the big effort invested, we were only able to prove here receptor activation but not receptor coupling. The transfection without G15 is a necessary control to exclude Gq endogenous coupling since both G alpha subunits, Galpha15 and Galphaq give intracellular increases in calcium levels. 

In the light of the results obtained, we wonder if the lack of productive interaction between cestode receptor and mammalian G alpha i/o subunit means that this interaction is not conserved or is divergent in Platyhelminths. This latter hypothesis far from uninteresting could have interesting implications from the perspective of parasite pharmacology. 

l.760: skip ‘…(neurotransmitters…others)…’

Removed as suggested.

l.761: Histamine also leads to an increase of the Ca++ dependent signal. Please explain!

We observed a very slight increase over basal levels in the Ca++ dependent signal in the presence of histamine at 10 µM which was lower than that observed with serotonin, which was 3 fold times higher than basal levels at serotonin concentrations of 10 µM. As can been seen in the new data (Fig 3C), the responsiveness to histamine is also seen in control HEK293-GNA15 cells, consistent with the known expression of histamine responses in the HEK293-GNA15 cell line. 

l.762ff: whole para is not convincing as outlined in my general comments. The authors have to address the ability of Eca-5-HT1a to inhibit cAMP production by a series of novel experiments.

The experiments suggested were performed and explained above but no inhibition in cAMP production could be detected. The experiments performed with the promiscuous G protein only suggest receptor activation but don´t give cues regarding coupling specificity. Receptor classification is only suggested here based on receptor´s phylogenetic and structural data and this was clearly explained in the discussion section. The lack of interaction between cestode receptor and mammalian G alpha protein, far from uninteresting, could mean a novelty in the way that Platyhelminths transduce signals and could have implications in the Parasite Pharmacology.

L768f: Misleading statement! So far, the authors have only in-appropriately examined coupling of Eca-5-HT1a to other G-proteins. As it reads, even in E. canadensis, the receptor would require G15 as an interacting partner to become functional.

We have removed the whole paragraph as suggested to avoid confusion. As we already explained previously in this response, coupling was analyzed by the downward cADDis system without success. As it was clearly stated in the discussion section (please see lines 1330 to 1335), coupling specificity is only suggested here based on receptor´s phylogenetic and modeling data. 

l.775-781: rephrase and skip all informations already given in Mat. and Meth.

Removed to avoid redundancy with the information given in Math & Meth and rephrased in lines 1236-1240.

l.800: ‘…shown in F…’

Changed as suggested, line 1271.

l.807: ‘…shown in H…’

Changed as suggested, line 1278.

l.836f: change to ‘…receptor-containing fibers could receive serotonergic…’

Changed as suggested, line 1308.

l.847: change to: ‘…assignment the proteins as a 5-HT1 type receptors…’

Change as suggested, line 1331.

l.851: skip info about spacing between TMs

Removed as suggested, line 1335.

l.858: change to ‘…is expected to be coupled…’

Change as suggested, line 1336.

l.862ff: statement concerning coupling to Gi-proteins not sufficiently examined (s.a.); skip

As we have explained previously, we have examined this issue using the downward cADDis system which couples the cAMP decreases with increases in fluorescence. Anyway, we have followed the review suggestion and the entire paragraph was removed as suggested, line 1339.

l.887-910: whole para suffers from the in-appropriately executed experiments! Conclusions are not supported by current data.

Thank you for this suggestion. We believe that we were unclear in our previous submission and the cADDis experiment was not properly described previously. Again, we have used the downward cADDis system which couples the cAMP decreases with increases in fluorescence. To detect cAMP decreases, the basal levels of cAMP were first increased with the constitutive Galphas mutant protein. Functional studies performed with the promiscuous Galpha15 protein only demonstrate activation of the receptor but not coupling. As it was clearly stated in the discussion section (please see lines 1330 to 1335), the evidence for the presumptive assignment the proteins as a 5-HT1 type of receptors are 1) phylogenetic analysis, 2) presence of residues in the ligand-binding pocket that are characteristic of 5-HT1 type, 3) presence of putative G-protein coupling residues (selectivity determinants of the receptor) that are more similar to the 5-HT1 type than to other clades, and finally, 4) structural features typical of 5-HT1 type of receptors. 

l.916: change to: ‘…images shown…to release serotonin that could be…’

Changed as suggested, please see lines 1463 and 1465.

l.923: change to: ‘…low titers and high concentrations of the serotonin…’

Changed as suggested, line 1470.

l.936: change to: ‘…Sm-5-HTR was expressed…’

Changed as suggested, line 1483.

l.946f: skip ‘…, a serotonergic receptor…[57].’

Section deleted as suggested, line 1502.

l.951-956: shorten para because until now UCM2550 has not been proven to be active at Eca-5-HT1a.

All the paragraph was deleted as suggested, line 1503.

l.961: change to ‘…receptor transcripts…’

Changed as suggested, line 1508.

l.963: change to: ‘…expression in the…stage is interesting…’

Changed as suggested, line 1510.

l.970ff: rephrase last sentence – strengthen wording

Rephrase as suggested, lines 1517 to 1569.

l.980: change to ‘…repurposing any of the approved drugs…’

Change as suggested, line 1577.

l.1344: change to ‘…GPCR genes…’; most likely heading will be no more required when moving receptor names and accession numbers to legend of Fig.1.

Removed as suggested.

l.1266: change to ‘Comparison of gene expression levels, determined…’

Change as suggested, line 1956.

Figure S2: skip labelling of N-glycosylation sites in ICL3

The labelling of N-glycosylation sites was removed from the figure as suggested.

Figure S4: skip completely

Removed as suggested.

Figure S5: skip completely; seriously, part D does not fit to the data shown in the main text!

Removed as suggested.

Figure S6: skip completely; as mentioned above, the data were obtained under in-appropriate conditions

Removed as suggested.

Reviewer #2: This manuscript describes the detailed structural analysis and functional characterisation of a new type of cestode 5HT receptor. Multiple different methods were used to characterise the receptor including in silico analysis, expression in HEK cells and immunolocalization. Authors concluded that the 5HT receptor is likely be classified as type 1.

The real strengths of this manuscript lie in the comprehensive structural analysis of the receptors identified with very strong data indicating type 1 characterisation. The authors have appropriately addressed technical difficulties common with functional expression of receptors in HEK cells and provided more than sufficient data to prove that the receptor functions as a serotonin receptor. Although authors could not definitively functionally characterise this 5HT receptor as type 1, this is reflected appropriately in the final conclusion of the study and the language used when describing the receptor. The results have been condensed and presented in a very clear format highlighting the most important data within the study. Immunolocalization images are easy to interpret showing appropriate expression patterns for the conclusions which have been drawn from the study. Although background staining using the serotonin antibody is obvious, the authors have addressed this issue within the manuscript/review process and the staining presented for the 5HT receptor antibody is very clear and much more defined. Combined serotonin and 5HT receptor staining is acceptable for the conclusions drawn using this data.

Overall I believe this manuscript should be accepted for publication as the authors have made an impressive effort to address all issues/concerns from reviewers and really focussed the details of the study. The data presented are accurate supporting the discussions/conclusions and the manuscript has been edited to a very high standard.

6. PLOS authors have the option to publish the peer review history of their article (what does this mean?). If published, this will include your full peer review and any attached files.

Do you want your identity to be public for this peer review? For information about this choice, including consent withdrawal, please see our Privacy Policy.

Reviewer #1: No

Reviewer #2: No

---

## [Editor Report · Decision Letter 1]

13 Oct 2021

Characterization of a new type of neuronal 5-HT G-protein coupled receptor in the cestode nervous system

PONE-D-21-12202R1

Dear Dr. Camicia,

We’re pleased to inform you that your manuscript has been judged scientifically suitable for publication and will be formally accepted for publication once it meets all outstanding technical requirements.

Kind regards,

Wolfgang Blenau

Academic Editor

PLOS ONE

Additional Editor Comments (optional):

Lines 540-542, ”The nucleotide 541 sequence from this E. canadensis cDNA was 1638 base pairs long with an open reading frame of 1638 bp”: Is the value 1638 bp correct for both, cDNA length and ORF? Please check!

Line 786, “concetration”: Please correct typo!
---

## [Editor Report · Acceptance letter]

28 Oct 2021

PONE-D-21-12202R1 

­­Characterization of a new type of neuronal 5-HT G- protein coupled receptor in the cestode nervous system 

Dear Dr. Camicia:

I'm pleased to inform you that your manuscript has been deemed suitable for publication in PLOS ONE. Congratulations! Your manuscript is now with our production department. 

Kind regards, 

on behalf of

Dr. Wolfgang Blenau 

Academic Editor

PLOS ONE